# Proteomics reveals signal peptide features determining the client specificity in human TRAP-dependent ER protein import

Duy Nguyen[1], Regine Stutz[2], Stefan Schorr[2], Sven Lang[2], Stefan Pfeffer[3], Hudson H. Freeze[4], Friedrich Förster[5], Volkhard Helms [1], Johanna Dudek[2] & Richard Zimmermann[2]

In mammalian cells, one-third of all polypeptides are transported into or across the ER membrane via the Sec61 channel. While the Sec61 complex facilitates translocation of all polypeptides with amino-terminal signal peptides (SP) or transmembrane helices, the Sec61-auxiliary translocon-associated protein (TRAP) complex supports translocation of only a subset of precursors. To characterize determinants of TRAP substrate specificity, we here systematically identify TRAP-dependent precursors by analyzing cellular protein abundance changes upon TRAP depletion using quantitative label-free proteomics. The results are validated in independent experiments by western blotting, quantitative RT-PCR, and complementation analysis. The SPs of TRAP clients exhibit above-average glycine-plus-proline content and below-average hydrophobicity as distinguishing features. Thus, TRAP may act as SP receptor on the ER membrane's cytosolic face, recognizing precursor polypeptides with SPs of high glycine-plus-proline content and/or low hydrophobicity, and triggering substrate-specific opening of the Sec61 channel through interactions with the ER-lumenal hinge of Sec61α.

---

[1] Center for Bioinformatics, Saarland University, 66041 Saarbrücken, Germany. [2] Medical Biochemistry and Molecular Biology, Saarland University, 66421 Homburg, Germany. [3] Max-Planck Institute of Biochemistry, Department of Molecular Structural Biology, 82152 Martinsried, Germany. [4] Sanford-Burnham-Prebys Medical Discovery Institute, La Jolla, CA 92037, USA. [5] Bijvoet Center for Biomolecular Research, Utrecht University, 3584 CHUtrecht, The Netherlands. These authors contributed equally: Duy Nguyen, Regine Stutz. Correspondence and requests for materials should be addressed to V.H. (email: volkhard.helms@bioinformatik.uni-saarland.de) or to J.D. (email: johanna.dudek@uks.eu) or to R.Z. (email: richard.zimmermann@uks.eu)

n human cells, the endoplasmic reticulum (ER) membrane is a major site for membrane protein biogenesis and the entry point into compartments of the endocytic and exocytic pathways for most soluble proteins[1–5]. Protein transport into the mammalian ER involves various transport components, and precursor polypeptides having amino-terminal signal peptides (SP) or amino-terminal transmembrane helices (TMH)[6,7]. In the cotranslational transport pathway, the ribonucleoprotein signal recognition particle (SRP) recognizes SP and TMH of nascent precursor polypeptides emerging from cytosolic ribosomes, and the resulting SRP/ribosome/nascent chain complex is targeted to the ER membrane by an SRP receptor (SR)[8,9]. The precursor polypeptides are then inserted into the Sec61-complex, i.e. the polypeptide-conducting channel of the ER membrane (Fig. 1a)[10–15]. This initial insertion can occur spontaneously or may involve substrate-specific auxiliary components[11,15–17], such

as the translocon-associated protein (TRAP) complex[16–28]. However, TRAP function and mechanism as well as its rules of engagement remained largely unknown.

The TRAP-complex was originally termed the signal-sequence receptor (SSR) complex[18–20]. It has been crosslinked to nascent polypeptides at late translocation stages[19,23] and has been demonstrated to physically associate with Sec61[20–22]. The ribosome-associated Sec61-complex and the TRAP-complex form a stable stoichiometric super-complex called a translocon[25–27]. In vitro transport studies showed that the TRAP-complex stimulates protein translocation depending on the efficiency of the SP in transport initiation[17]; Sec61 gating efficiency and TRAP dependence were inversely correlated. Recent studies in intact cells suggest that TRAP may also affect TMH topology[28]. Furthermore, mutations in the human TRAPγ and TRAPδ subunits (SSR3 and SSR4, respectively) result in loss of TRAP and

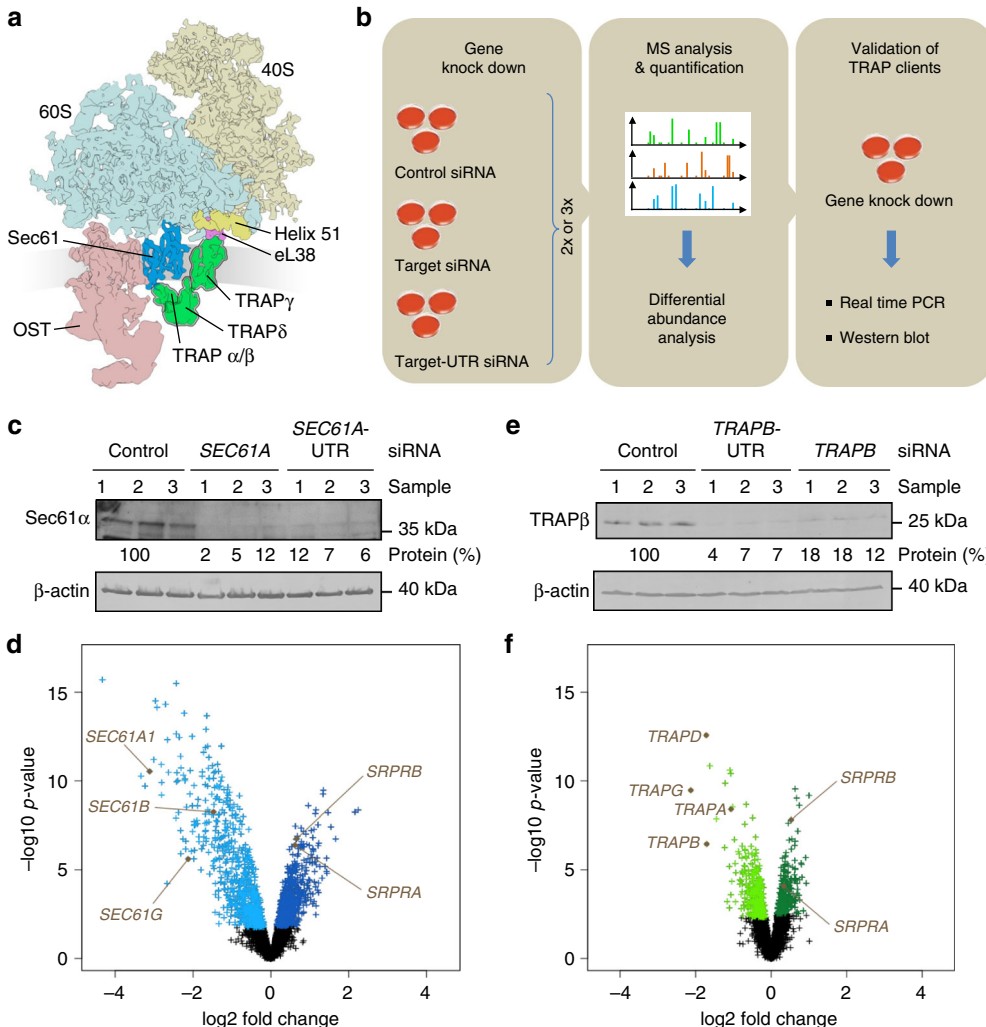

**Fig. 1** Identification of TRAP clients and compensatory proteins by TRAP depletion in HeLa cells. **a** Cartoon of clipped 80S ribosome together with Sec61-complex (blue color code in subsequent panels and figures), TRAP-complex (green color code in subsequent panels and figures), and OST[27]. Without clipping, eL38 and helix 51 would be partially hidden. **b** The experimental strategy was as follows: siRNA-mediated gene silencing using two different siRNAs for each target and one non-targeting (control) siRNA, respectively with six/nine replicates for each siRNA in two/three independent experiments; label-free quantitative proteomic analysis; and differential protein abundance analysis to identify negatively affected proteins (i.e., clients) and positively affected proteins (i.e. compensatory mechanisms). **c**, **e** Knock-down efficiencies in experiment 1 were evaluated by western blot. Results are presented as % of residual protein levels (normalized to ß-actin) relative to control, which was set to 100%. Blot results for other experiments are shown in Supplementary Fig. 1a, d. **d**, **f** Differentially affected proteins were characterized by the mean difference of their intensities plotted against the respective permutation false discovery rate-adjusted p-values in volcano plots (n = 2 in the case of Sec61 depletion, n = 3 in the case of TRAP depletion). The results for a single siRNA are shown in each case (SEC61A1-UTR siRNA, TRAPB siRNA). Additional plots are shown in Supplementary Fig. 1c, f. Subunits of the Sec61- and TRAP-complexes and of the SRP receptor are indicated

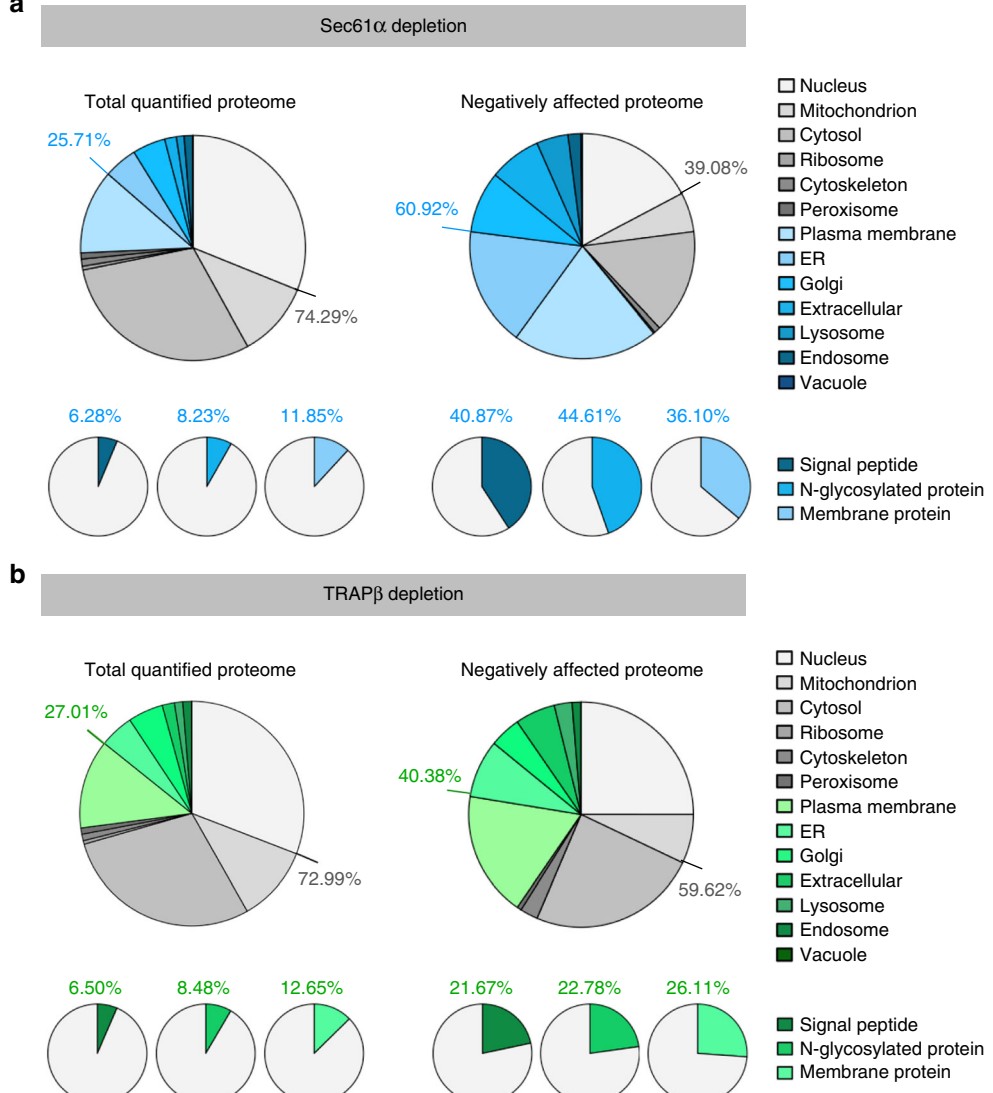

**Fig. 2** Validation of TRAP clients based on Gene Ontology enrichment factors. The color coding follows Fig. 1. Protein annotations of signal peptides, membrane location, and N-glycosylation in humans were extracted from UniProtKB, and used to determine the enrichment of Gene Ontology annotations among the secondarily affected proteins. **a**, **b** Summaries of two Sec61 (**a**) and three TRAP depletion experiments (**b**), performed in triplicate in each case

congenital disorders of glycosylation (CDG), suggesting that TRAP plays a direct or indirect role in protein N-glycosylation[29,30].

Traditionally, the substrate specificities of mammalian protein transport components (e.g., the TRAP-complex) have been investigated in cell-free translation reactions in which a small set of (artificial) model precursor proteins is synthesized one-by-one in the presence of reconstituted ER membranes[17], or in pulse-chase experiments in human cells that overproduce the model precursor[28]. These approaches are suitable for addressing whether a certain component can stimulate ER import of a given precursor polypeptide. However, due to the bias of these experimental strategies, they fail to clearly define the characteristics of precursor polypeptides that lead to TRAP dependence.

Here, we identify and characterize the native precursor polypeptides that depend on TRAP in human cells under physiological conditions. To this end, we combine siRNA-mediated gene knock-down in HeLa cells with label-free quantitative proteomic analysis and differential protein abundance analysis (Fig. 1b). SP analysis of the TRAP clients reveals above-average glycine-plus-proline content as the distinguishing feature for TRAP

dependence and, thus, suggests an hitherto undetected SP heterogeneity. We propose that this SP heterogeneity may provide an opportunity for regulation of transport of a subset of precursor polypeptides and may be linked to both TRAP mechanism and CDG etiology.

## Results

**The client specificity of Sec61 in human ER protein import**. As a proof of concept, HeLa cells were depleted of the Sec61-complex using two different *SEC61A1*-targeting siRNAs[15]. We assessed the proteomic consequences of this knock-down via label-free quantitative proteomics[31] and differential protein abundance analysis relative to cells treated with non-targeting (termed control) siRNA (Fig. 1b). It was previously established that the utilized gene silencing technique leads to >90% depletion of the Sec61-complex, without substantially affecting cell growth, cell viability, or cell/ER morphology[15]. We confirmed the silencing efficiency by western blot (Fig. 1c, Supplementary Fig. 1a).

After Sec61 depletion, we quantitatively characterized 7212 ± 356 different proteins (mean value with standard deviation, $n = 2$) by mass spectrometry (MS), representing roughly 50%

of the cellular proteome (see MS proteomics data with identifier PXD008178 (https://www.ebi.ac.uk/pride/archive/projects/PXD008178)). Of these proteins, 5129 were detected in all samples and were therefore statistically analyzed (Supplementary Data 1). They included good representation of proteins with cleaved SP (6%), glycoproteins (8%), and membrane proteins (12%) (Fig. 2a, Supplementary Fig. 1b)[31,32]. Statistical analysis of the ratio changes after targeting versus non-targeting siRNA treatment ($q < 0.05$, i.e. permutation false discovery rate-adjusted $p$-value) revealed that Sec61α depletion significantly affected the steady-state levels of 824 proteins: 482 negatively and 342 positively (Fig. 1d, Supplementary Fig. 1c, Supplementary Data 3, 4). As expected[15], Sec61α itself was negatively affected (Fig. 1d, Supplementary Fig. 1c). The proteomic approach confirmed that the Sec61β and Sec61γ subunits were degraded upon depletion of Sec61α[15]. Among the other negatively affected proteins, Gene Ontology (GO) terms assigned 61% to organelles of the endocytic and exocytic pathways, representing a strong enrichment compared to the value for the total quantified proteome (26%) (Fig. 2a). We also detected significant enrichment of precursor proteins with SP (6.5-fold), N-glycosylated proteins (5.4-fold), and membrane proteins (3.0-fold) (Fig. 2a). This suggests that the precursors of these proteins, 197 with SP and 98 with TMH, are substrates of the Sec61-complex and were therefore degraded by the proteasome upon its depletion. Also as expected[15], the positively affected proteins included compensatory components, including the two subunits of the SRP receptor (Fig. 1d, Supplementary Fig. 1c, see paragraph on potential compensatory mechanisms). Bioinformatic analysis predicts that ~30% of the total quantified proteome comprises Sec61 substrates. Thus, our experimental approach underestimated the number of different precursor polypeptides that rely on the Sec61-complex. This may be explained by some precursors having a longer half-life compared to Sec61 or a higher than average affinity for Sec61 (see paragraph on characteristics of TRAP clients).

In summary, our experimental strategy in human cells was successfully used to analyze the client spectrum of the Sec61-complex—an essential transport component. These results further set the stage for subsequent analysis of precursor-specific transport components, such as the TRAP-complex.

**The client specificity of TRAP in human ER protein import.** We next performed similar analyses after TRAP-complex depletion using two different *TRAPB*-targeting siRNAs in comparison to non-targeting (termed control) siRNA. It was previously established that this gene silencing method resulted in >90% TRAP-complex depletion, without significantly affecting cell growth, cell viability, or cell/ER morphology[26]. We confirmed silencing efficiency by western blot (Fig. 1e, Supplementary Fig. 1d).

After TRAP depletion, $7670 \pm 332$ different proteins (mean value with standard deviation, $n = 3$) were quantitatively characterized by MS, 5911 of which were detected in all samples (Fig. 2b, Supplementary Fig. 1e, Supplementary Data 2). Notably, the observed difference of about 460 total proteins to the *SEC61A1* silencing experiments is not statistically significant. Applying the same statistical analysis as used for *SEC61A1* silencing, we found that TRAPβ depletion significantly affected the steady-state levels of 257 proteins: 180 negatively and 77 positively ($q < 0.05$) (Fig. 1f, Supplementary Fig. 1f, Supplementary Data 5, 6). As expected[26], TRAPβ itself was negatively affected (Fig. 1f, Supplementary Fig. 1f). Proteomic analysis confirmed that TRAPβ depletion was accompanied by degradation of TRAPα, TRAPδ, and TRAPγ[26]. Of the other negatively affected proteins, GO terms assigned ~40% to organelles of the

endocytic and exocytic pathways (Fig. 2b). We also detected significant enrichment of proteins with SP (3.3-fold), N-glycosylated proteins (2.7-fold), and membrane proteins (2.1-fold) (Fig. 2b). Among the proteins negatively affected upon TRAPβ depletion, the precursors are potential clients of the TRAP-complex and were most likely degraded by the proteasome upon its depletion (see paragraph on validation of TRAP client characteristics). The identified precursors included 38 proteins with cleavable SP and 22 membrane proteins, and represented N-glycosylated proteins and non-glycosylated proteins (Fig. 2b, Tables 1 and 2, Supplementary Data 5). The fact that the numbers of negatively affected proteins after TRAP depletion are lower as compared to the negatively affected proteins after Sec61 depletion is consistent with TRAP being a precursor-specific auxiliary transport component to the Sec61-complex. Interestingly, only 40% of the potential TRAP clients were also negatively affected by Sec61 depletion (Tables 1 and 2, Supplementary Fig. 2a). We attribute this to the observation that the efficiencies, with which SPs gate Sec61-channels, were inversely correlated with their TRAP dependence[17]. Alternatively, TRAP dependent and seemingly Sec61 independent substrates may have a higher than average affinity for Sec61-TRAP-super-complexes and thus may have used the residual super-complexes more efficiently than many TRAP independent substrates. The positively affected proteins included the SRP receptor subunits (Fig. 1f, Supplementary Fig. 1f, see the following paragraph on potential compensatory mechanisms).

**Potential compensatory mechanisms after TRAP depletion.** To investigate potential compensatory mechanisms of TRAP-complex depletion, independent silencing experiments were subjected to quantitative RT-PCR and western blotting for the two SRP receptor subunits, which were among the positively affected proteins (Fig. 1d, f, Supplementary Fig. 1c, f, Supplementary Data 6, 7). Western blot with independent samples confirmed elevated levels of these subunits, and quantitative RT-PCR revealed that these increases resulted from increased protein synthesis or stability rather than increased transcription of the respective genes (Supplementary Fig. 3). These results are in line with our previous observation that these two proteins are present at higher concentrations after depletion of other transport components, such as Sec61α and Sec62[15,33]. Additional proteins that were positively affected by both depletions were the two cytosolic proteins helicase-like transcription factor (HLTF) and Midline 1 (MID1) (Supplementary Data 4, 6, 7). Interestingly, both proteins have ubiquitin-ligase activity and were previously connected to cell migration and collagen biogenesis, respectively[34,35]. There was no indication for activation of the unfolded protein response (UPR) in the course of the 96 h knock-down, i.e., related terms did not come up as enriched GO terms in the analysis of the positively affected proteins and typical UPR-regulated genes such as *HSPA5, HSPB1*, and *HYOU1* were not up-regulated (Supplementary Data 7, see the following paragraph on validation of TRAP clients).

In case of Sec61 depletion, many more E3 ubiquitin-ligases were up-regulated, amounting to a total of 11 of the 330 positively affected proteins as were eight cytosolic molecular chaperones, both being consistent with the cytosolic accumulation of precursor polypeptides in the absence of Sec61 complex (Supplementary Data 4). Furthermore, under these conditions additional protein targeting components were up-regulated, too, such as subunits of the ER targeting components SRP (SRP68, SRP54) plus TRC receptor (GET4) and the mitochondrial protein receptor and import complexes TOM (TOM6, TOMM7) and TIM (TIMM23), both being in line with our previous

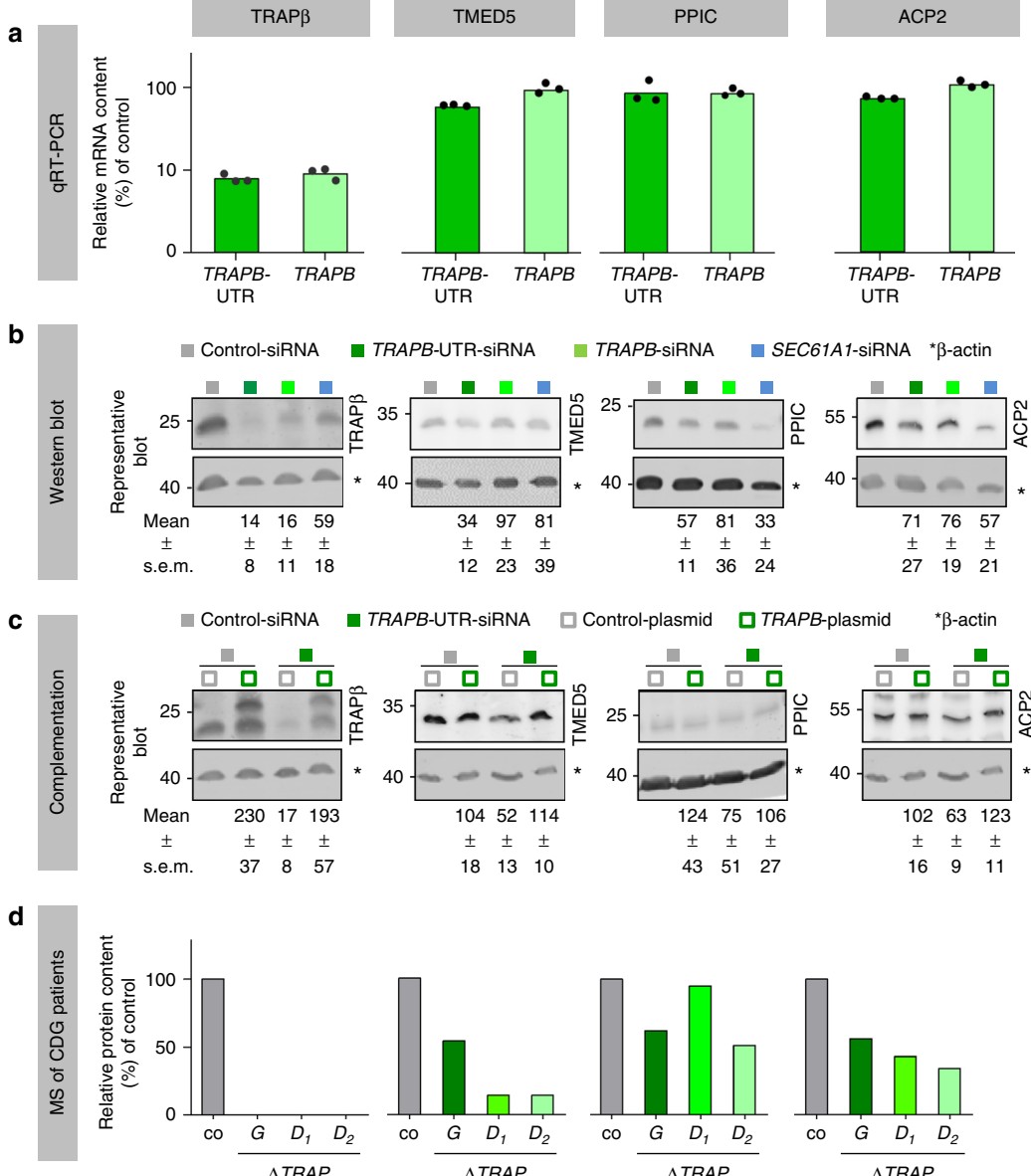

**Fig. 3** Validation of TRAP clients by western blot and quantitative RT-PCR. The color coding follows Fig. 1. **a–c** HeLa cells were depleted of TRAP- or Sec61-complex using two different *TRAPB*-targeting siRNAs, or *SEC61A1*-UTR-targeting siRNA, or treated with a non-targeting (control) siRNA, and the consequences of complex depletion were analyzed by quantitative RT-PCR and western blots for TRAPβ and TRAP client candidates. **a** Quantitative RT-PCR data represent the mean mRNA values relative to control and the corresponding dot plots for nine replicates for each siRNA from three independent experiments. **b**, **c** Quantitative western blot data represent the mean protein levels (normalized to ß-actin) relative to control and standard errors of the mean (s.e.m.) for five to six (**b**) or three (**c**) independent experiments. **c** Silencing phenotypes were rescued by the indicated complementation and analyzed by western blot. In the case of TRAPβ, the upper band represents the tagged protein and the numbers refer to the sum of tagged and un-tagged protein. **d** Control fibroblasts (co) as well as TRAP-deficient fibroblasts from CDG patients with mutations in *TRAPG* (G, one patient) or *TRAPD* (D, two patients) genes[27] were analyzed by quantitative proteomics

observations that protein targeting pathways to the ER have overlapping specificities[36,37] and that some ER targeted precursor polypeptides enter mitochondria in the absence of proper ER targeting[38]. Overall, these results are consistent with the view that Sec61 depletion for 96 h had a more severe impact on the capacity of the secretory pathway as compared to TRAP depletion, i.e. that TRAP serves as a precursor-specific auxiliary transport component to the Sec61-complex.

**Validation of TRAP clients**. To validate the TRAP clients, we conducted independent silencing experiments with *TRAPB*-

targeting siRNAs, and subjected three SP-containing candidates representing various precursor types to quantitative RT-PCR and western blotting (ACP2, PPIC, TMED5). Cells treated with non-targeting siRNA or *SEC61A1*-targeting siRNA served as negative and positive controls. Comparison of qRT-PCR and western blot data confirmed that *TRAPB* siRNA-treated HeLa cells exhibited simultaneous depletion of *TRAPβ* mRNA and TRAPβ protein, and concomitant TRAPα depletion without disruption of *TRAPα* mRNA levels. Similar analyses of the three analyzed client candidates confirmed that the proteins were depleted with little change in the corresponding mRNAs (Fig. 3a, b, Supplementary Fig. 3).

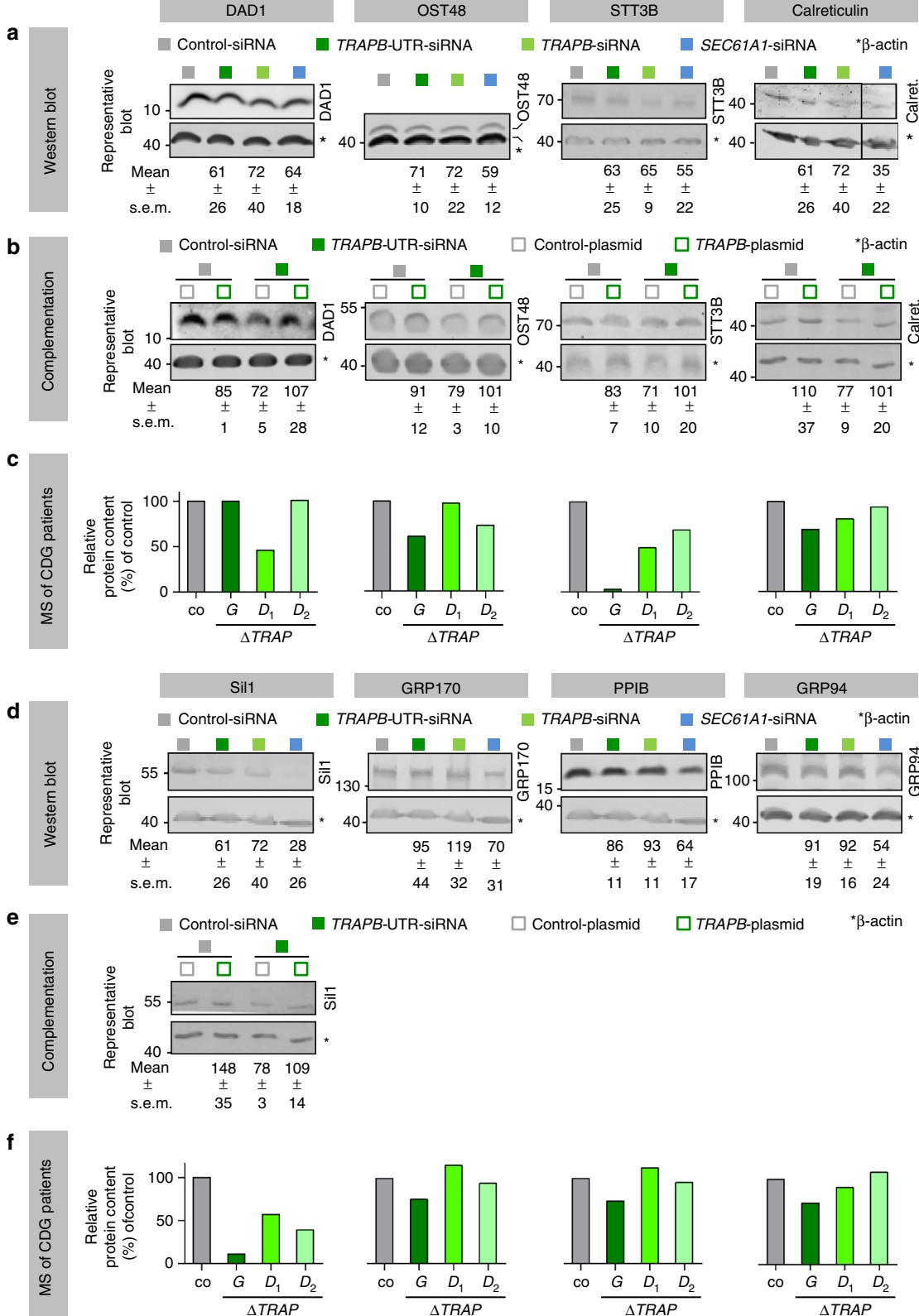

**Fig. 4** Validation of TRAP clients by western blot and proteomic analysis of CDG patients. **a**, **b**, **d**, **e** HeLa cells were treated with two different *TRAPB*-targeting siRNAs, or *SEC61A1*-UTR-targeting siRNA, or control siRNA, and the consequences of complex depletion were analyzed by western blots for TRAP client candidates (**a**), which were deduced from the proteomic analysis, possible TRAP clients (**a**, **d**), and negative control proteins (**d**). Where indicated, silencing phenotypes were rescued by the indicated complementation and analyzed by western blots (**b**, **e**). Quantitative western blot data represent the mean protein levels (normalized to ß-actin) relative to control and standard errors of the mean (s.e.m.). Western blots refer to at least three independent experiments. **c**, **f** In addition, TRAP-deficient fibroblasts from CDG patients with mutations in *TRAPG* or *TRAPD* genes[27] were analyzed by quantitative proteomics

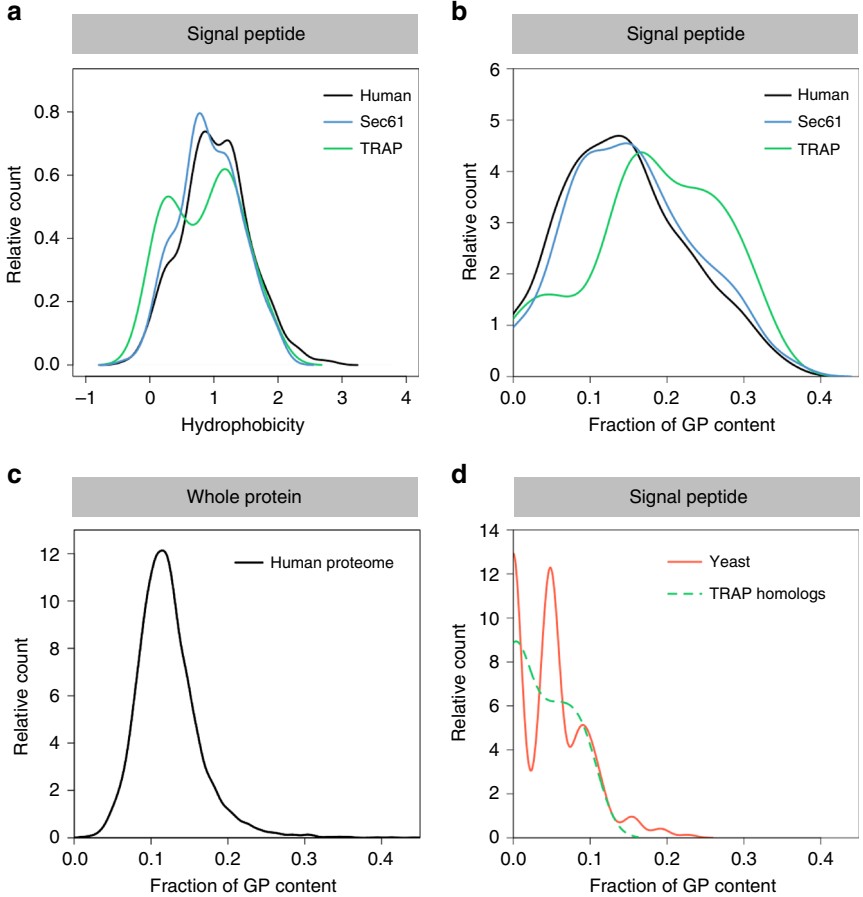

**Fig. 5** Physicochemical properties of TRAP clients with SP. The color coding follows Fig. 1. We used custom scripts to compute the hydrophobicity score (**a**) and glycine/proline (GP) content (**b**) of SP sequences. Hydrophobicity score was calculated as the averaged hydrophobicity of its amino acids according to the well-known Kyte-Doolittle propensity scale. GP content was calculated as the total fraction of glycine and proline in the respective sequence. Additional plots are shown in Supplementary Fig. 4a, b. We also used custom scripts to extract protein annotations for all human proteins (**c**) and yeast SP (**d**) from UniProtKB entries. TRAP homologs refer to yeast orthologs of eight TRAP dependent human proteins as indicated in Table 1. **d** Notably, the first peak refers to no GP per yeast SP, the second to one GP, the third to two GPs, and so on. This oscillatory appearance appears to be obscured by the higher variation in length of human SP

For further validation, we performed complementation analysis. Simultaneous transfection of cells with *TRAPB*-targeting siRNA and a plasmid allowing expression of siRNA-resistant *TRAPB*-cDNA rescued depletion of both the TRAP-complex and the potential TRAP clients (Fig. 3c, Supplementary Fig. 3c). Thus, the observed effects were considered specific, and the three candidates were characterized as true TRAP clients.

Mutations in the human TRAPγ and TRAPδ subunits reportedly result in partial or complete loss of TRAP-complex, which leads to CDG[27,29,30]. Therefore, we also used quantitative mass spectrometry to analyze the potential TRAP candidates in CDG patient fibroblasts for candidate evaluation (Fig. 3d). The results from these chronically TRAP-depleted cells confirmed the findings of the experiments in which HeLa cells were acutely depleted of TRAP-complex using *TRAPB*-targeting siRNA.

To validate further TRAP clients, we conducted independent silencing and western blot experiments with *TRAPB*-targeting siRNAs and complementation assays for three more candidates according to Tables 1 and 2, representing OST subunits (Dad1, Ost48, Stt3b), and for two potential candidates (Sil1, Calreticulin), which had been suggested by the proteomic analysis by only one of the two siRNAs, i.e. had not passed the significance threshold ($q < 0.05$). Furthermore, we subjected three SP-containing proteins (GRP94, GRP170, PPIB), which had been affected by Sec61 depletion but not by TRAP depletion, to the same analysis.

Again, cells treated with non-targeting siRNA or *SEC61A1*-targeting siRNA served as controls. Both western blot experiments and complementation assays confirmed the three OST subunits plus the two potential TRAP clients as true TRAP clients (Fig. 4a, b, d, e). In contrast, the three additionally analyzed proteins were not characterized as TRAP clients (Fig. 4d). The quantitative mass spectrometry of all these proteins in CDG patient fibroblasts was consistent with this interpretation (Fig. 4c, f).

Overall, our results confirmed that the experimental approach of siRNA-mediated TRAP knock-down, label-free quantitative proteomic analysis, and differential protein abundance analysis had successfully identified true TRAP client precursor polypeptides.

**Characteristics of TRAP clients**. We next analyzed the Sec61 and TRAP clients with respect to the physico-chemical properties of their amino-terminal SP and TMH. Precursors with less-hydrophobic SP were more strongly affected by Sec61 depletion, i.e., over-represented in the affected polypeptides, suggesting that Sec61 prefers precursor polypeptides with a higher-than-average SP hydrophobicity (Fig. 5a) (Wilcoxon rank test *p*-value of 0.055). As a distinguishing feature for TRAP dependence, we observed a tendency towards a lower overall SP hydrophobicity

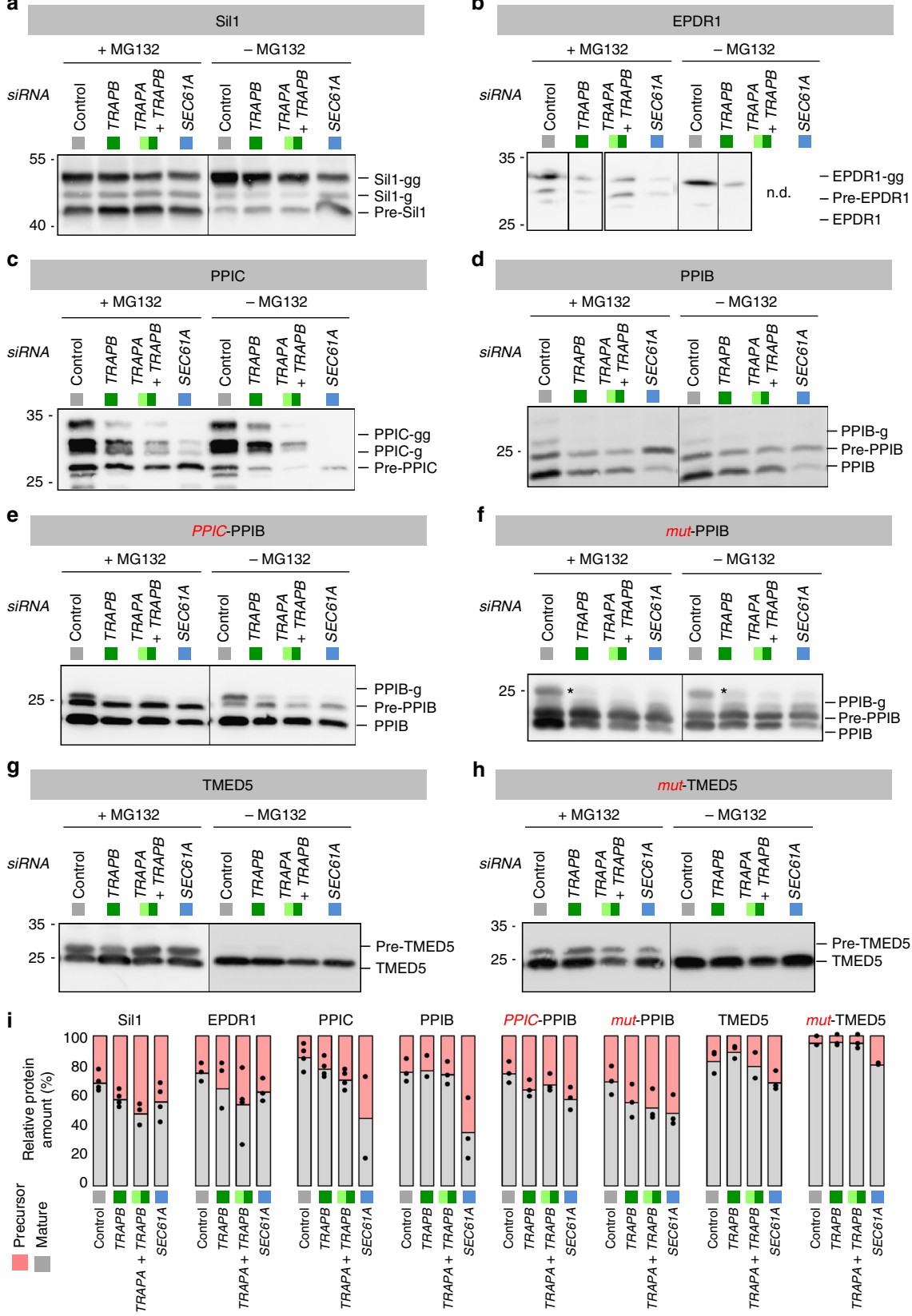

**Fig. 6** Validation of physicochemical properties of TRAP clients. **a–h** Plasmid driven over-production of model precursor polypeptides in HeLa cells was carried out in the presence of the indicated siRNA for 24 h. Where indicated, MG 132 was present during the last 8 h. Samples were analyzed by western blot. The identity of the mature proteins is based on experiments where N-glycosylation was inhibited by Tunicamycin (Supplementary Fig. 6a–h). We note that the pre-form represents the precursor polypeptide, the form without any addition the mature protein without N-glycosylation, g the mono-gylcosylated mature protein, and gg the doubly-glycosylated mature protein. **i** In case of the experiments with MG 132, quantitative western blot data represent the mean precursor and mature protein (N-glycosylated plus non-glycosylated) levels plus the corresponding dot plots for at least three independent experiments. The western blots for the depleted proteins are shown in Supplementary Fig. 6i. *, unspecific band

(Fig. 5a) ($p = 0.125$), which became significant ($p = 0.05$) when the analysis was confined to proteins which were also affected by Sec61 depletion (Supplementary Fig. 4a). This is consistent with previous in vitro transport data[17]. More significantly, however, TRAP dependent SP showed a higher glycine-plus-proline (GP) content ($p = 0.007$), which may be indicative of a lower helix propensity (Fig. 5b, Table 1) and has not been previously reported[39]. Notably, this latter SP feature remained relevant ($p = 0.06$) when the analysis was confined to proteins which were also affected by Sec61 depletion (Supplementary Fig. 4b). The average GP content of TRAP dependent SP was increased by 50% as compared to all human SP, as well as all human proteins (Fig. 5b, c). Notably, the GP content of the SP of the two potential TRAP clients, Sil1 and Calreticulin, and the three TRAP independent proteins, GRP94, GRP170, and PPIB, was near and below, respectively, the 15% GP threshold (Table 3). Visual inspection of the summarized data also suggested lower overall hydrophobicity and higher GP content for TMH of TRAP dependent membrane proteins without cleavable SP (Supplementary Fig. 4c, d, Table 2). However, these associations did not demonstrate statistical significance.

To investigate the possible origin of the unusually high GP content in the SP of TRAP clients, we investigated homologs of human TRAP clients in *S. cerevisiae*. Among almost 7000 yeast protein sequences extracted from SwissProt, over 800 contain SP. Seven pairs of sequences exhibited SP in both the TRAP client and its *S. cerevisiae* homolog. In these cases, the SP of the *S. cerevisiae* homologs showed an average GP content of 5.3% (Fig. 5d). Furthermore, SP in *S. cerevisiae* generally showed a GP content of 8% (Fig. 5d versus 5b). Since yeast do not have TRAP, these findings support the relevance of high GP content for TRAP client translocation in human cells.

**Validation of TRAP client characteristics**. Four different approaches allowed us to address the relevance of the observed SP characteristics for TRAP dependence in ER protein import. In the first approach, we searched in our data set for protein paralogs, which are similar in amino acid sequence within their mature domains but have different GP content in their SP. The two peptidylprolyl-cis/trans-isomerases PPIB and PPIC represent such paralogs. They have 72% sequence identity plus another 13% similar amino acids within their mature domains (Supplementary Fig. 5a). PPIB has a GP content of 12% in its SP (Table 3), whereas PPIC has 32% (Table 1). According to the validation data in Figs. 3 and 4, the precursor of PPIC depends on TRAP for ER import, whereas PPIB does not. Thus nature has already done a SP swapping experiment and supports the notion that SP characteristics drive TRAP dependence and that a high GP content may play an important role in rendering a precursor polypeptide TRAP dependent (see Discussion).

Secondly, *myc*DDK-tagged variants of the same two PPI precursors plus the TRAP dependent model precursor proteins Sil1 and EPDR1 were overproduced for 24 h in HeLa cells, which were treated with non-targeting siRNA, or transfected with either *TRAPB*-, or a combination of *TRAPA* plus *TRAPB*, or *SEC61A1*-targeting siRNAs, in the absence or presence of the proteasome inhibitor MG132. According to western blot analysis, PPIB import was only affected by Sec61-complex depletion, whereas the precursors of PPIC, EPDR1, and Sil1 also accumulated in the absence of TRAP-complex when the proteasome was inhibited (Fig. 6a–d, i, Supplementary Fig. 6a–d). Thus, these short-term expression experiments are consistent with a crucial role of a high GP content in SP of TRAP dependent precursor polypeptides. Furthermore, these data experimentally demonstrated for three model precursor polypeptides that our starting assumption was correct: Precursor polypeptides are degraded by the proteasome when they are accumulating in the cytosol in the absence of Sec61- and/or TRAP-complex.

Thirdly, we created by molecular cloning a hybrid precursor comprising the GP-rich SP of a TRAP dependent precursor (PPIC) with the mature part of a TRAP independent precursor (PPIB). The hybrid precursor of this SP swapping experiment, termed *PPIC*-PPIB, phenocopied PPIC, i.e., accumulated in the absence of Sec61- as well as TRAP-complex in the short-term expression analysis (Fig. 6e, i, Supplementary Fig. 6e). This result supports our conclusion that TRAP dependence of precursor polypeptides in ER protein import is dominated by the signal peptide and that the SP GP content may play an important role.

To take the validation yet one step further, the GP-rich SP of a TRAP dependent precursor (TMED5) and the SP of a TRAP independent precursor (PPIB) were changed to the opposite GP values with as little as possible impact on SP hydrophobicity by quick change mutagenesis. The two mutated precursors, termed *mut*-TMED5 and *mut*-PPIB, respectively, decreased the GP content of the TMED5 SP from 29.6 to 22.2 and increased the GP of the PPIB SP from 12.1 to 24.2, while changing the hydrophobicity only from 1.074 to 1.250 and from 0.961 to 0.814 (Table 3). According to the protean prediction tool of the DNASTAR software (Lasergene 12), these mutations increased the helix propensity of the TMED5 SP and decreased the helix propensity of the PPIB SP (Supplementary Fig. 5b). These mutant variants were also subjected to the short term expression analysis. While the exchange of a PG pair for two alanines in the case of the TMED5 SP turned the TRAP dependent precursor into a TRAP independent one, the simultaneous replacements of the dipeptide SE by GP plus two alanines by the dipeptide PP in the PPIB SP had the opposite effect (Fig. 6f-i, Supplementary Fig. 6f-h). Thus, the combination of SP mutagenesis and short-term expression experiments strongly supports a crucial role of a high GP content in SP of TRAP dependent precursor polypeptides.

**Validation of clients and characteristics in CDG patients**. As an additional validation, we subjected two control fibroblasts and three CDG patient fibroblasts with TRAP-deficiency to label-free quantitative proteomic analysis and differential protein abundance analysis and analyzed the data for negatively affected proteins, i.e. potential TRAP clients (Supplementary Data 8). Notably, the same fibroblasts were previously used for cryo-electron tomography of the respective translocons in their native ER membranes[27]. Here, we quantitatively characterized a total of 5,919 different proteins by mass spectrometry, 279 of which were negatively affected by TRAP deficiency in the three patient

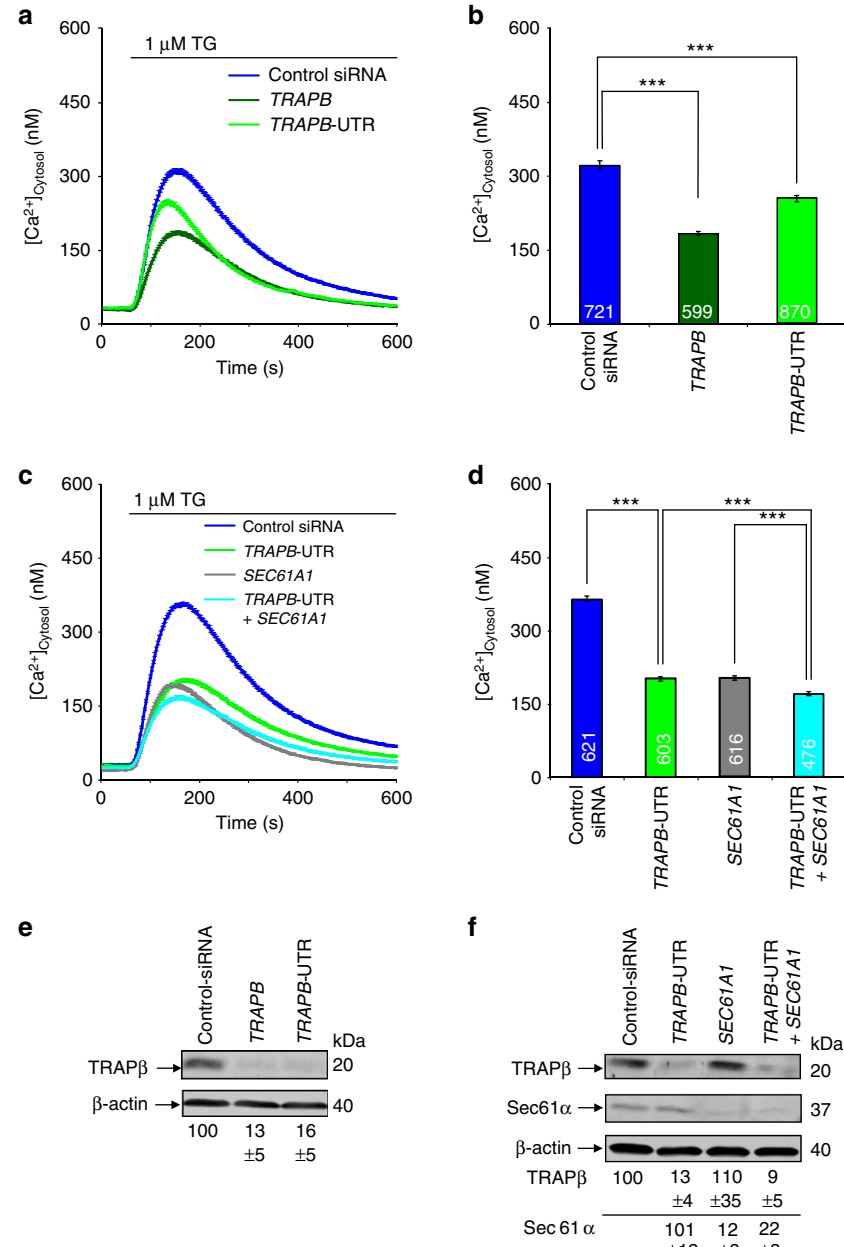

**Fig. 7** TRAP depletion plus live-cell Ca²⁺ imaging reveals a TRAP function in Sec61-channel opening. HeLa cells were treated with the indicated siRNAs for 96 h, loaded with Fura 2, and subjected to live-cell imaging of cytosolic Ca²⁺ following our established procedure. Ca²⁺ release was unmasked by the addition of thapsigargin (TG) in the presence of external EGTA. **a**, **c** Average values are presented. Error bars represent standard error of the mean (s.e.m.). **b**, **d** Statistical analysis of the changes in cytosolic Ca²⁺ after TG addition in **a**, **c**. Error bars represent s.e.m. *P*-values of <0.001 by unpaired *t*-test were defined as significant, and are indicated by three asterisks (***). The numbers of analyzed cells are indicated. Data were collected in at least three independent experiments, with triplicate cultures for each condition. Knock-down efficiency was evaluated by western blots. **e**, **f** A representative western blot is shown, along with the silencing statistics (mean values with s.e.m.)

fibroblasts versus control fibroblasts using the same analysis workflow as in Fig. 1d, f. Fifteen of these 279 proteins had also been negatively affected by TRAP depletion in HeLa cells. Proteomic analysis confirmed the almost complete absence of TRAP complex, as seen as absence of TRAPß, in fibroblasts from CDG patients with mutations in the *TRAPG* or *TRAPD* genes[27] (Supplementary Tables 1, 2). Furthermore, this analysis confirmed the absence of the OST subunit TUSC3 (Supplementary Table 1). 36% of the negatively affected proteins, i.e. 100 proteins were assigned to the secretory pathway, including 41 membrane proteins and 34 proteins with SP (Supplementary Fig. 7a, b). There was hardly any overlap between these proteins and the proteins

negatively affected by transient TRAP complex depletion in HeLa cells, i.e. none for membrane proteins and only four proteins with SP (not counting TRAP subunits) (Supplementary Fig. 2b). Strikingly, 30 of the negatively affected SP proteins are N-glycoproteins, 15 have SP with a GP content of >15% (Supplementary Fig. 7b-d, Supplementary Table 1). In the case of membrane proteins with TMH, 17 are N-glycoproteins, 9 have TMH with GP content of >15% (Supplementary Fig. 7b, e-f, Supplementary Table 2). Thus, the results from these chronically TRAP-depleted cells partially confirmed that the GP content of SP plays an important role for TRAP dependence of precursor polypeptides in ER protein import. However, the results from the TRAP

deficient patient fibroplasts were obviously more blurred by secondary effects than those from transiently depleted HeLa cells (Supplementary Fig. 7a, b versus Fig. 2b). In addition, these results confirmed the N-glycosylation deficiency, which was seen in the corresponding CDG patients, and suggested that this may result directly from the depletion of TRAP plus from its secondary effects on OST. Interestingly, the CDG patient analysis also confirmed the up-regulation of E3 ubiquitin-ligases (HERC2, TRIM4), as it had been observed for transiently TRAP depleted HeLa cells (Supplementary Data 9).

**TRAP affects Sec61-channel gating.** In vitro transport studies show that TRAP stimulates the initial insertion of nascent precursor polypeptides into the Sec61-channel, and exists in proximity to soluble and membrane protein precursors at late stages of their transit through the Sec61-channel[17,23]. Therefore, it was proposed that TRAP may facilitate opening of the Sec61- channel either through direct interaction, or by acting as a molecular ratchet on the incoming precursor polypeptide, or both. In its open state, the Sec61-channel also allows passive $Ca^{2+}$ efflux from the ER; therefore, Sec61-channel opening can be monitored in intact cells via live-cell $Ca^{2+}$ imaging[40,41]. Thus, in our present work, HeLa cells were depleted of TRAP-complex, and ER $Ca^{2+}$ leakage was monitored based on the increase of cytosolic $Ca^{2+}$. In contrast to treatment with control siRNA, cellular TRAP depletion with one of two different siRNAs directed against *TRAPB* resulted in decreased ER $Ca^{2+}$ leakage (Fig. 7a, b). With simultaneous depletion of Sec61-complex (which itself leads to reduced ER $Ca^{2+}$ efflux), *TRAPB* siRNA had only a slight effect (Fig. 7c, d). This suggests that the observed effect of TRAPβ depletion occurred at the level of the Sec61-channel. These results are consistent with the TRAP-complex acting in the opening of the Sec61 channel for protein translocation; however, they do not exclude the possibility that the TRAP-complex may additionally act as molecular ratchet on incoming polypeptide chains. Notably, our previous work did not detect any changes of sarcoplasmic/endoplasmic reticulum $Ca^{2+}$ ATPase (SERCA) after *SEC61A1* silencing for 96 h[15].

## Discussion
The Sec61-complex facilitates translocation of all polypeptides with amino-terminal signal peptides (SP) or amino-terminal transmembrane helices (TMH) into the ER[4,5]. The TRAP-complex supports translocation via the Sec61-channel in a substrate-specific manner[17]. To characterize TRAP dependent precursors, we combined siRNA-mediated TRAP depletion in human cells, label-free quantitative proteomics, and differential protein abundance analysis. By applying our unbiased approach in living human cells, we identified 60 potential TRAP clients that included precursors of soluble and membrane proteins with cleavable SP (38) and membrane proteins without cleavable SP (22), and precursors of both N-glycosylated and non-glycosylated proteins. Six of these potential clients (ACP2, PPIC, TMED5, Dad1, Ost48, Stt3b) plus two additional precursors (Calreticulin, Sil1), i.e., negatively affected proteins which had not passed the significance threshold, were confirmed as TRAP dependent in independent experiments by western blots and rescue experiments by *TRAPB* complementation. These client classes are consistent with previous findings[17,28]. Our identification of non-glycosylated TRAP clients suggests that TRAP dependence is not just an effect of TRAP playing a direct role in N-glycosylation. Closer inspection revealed that putative TRAP clients more commonly showed lower overall hydrophobicity and, most significantly, higher-than-average glycine-plus-proline content in their cleavable SP (and TMH). This includes four subunits of

oligosaccharyl transferase (OST), which have the highest potential to contribute to the phenotype of CDG patients. We note that the Calreticulin SP was present in the CDG reporter construct[29]. Of the roughly 3500 human SP in UniProtKB, almost 1400 have a GP content of >15%, and thus may represent TRAP clients, which may allow for regulated access of the respective precursor polypeptides to the Sec61-channel.

We have attempted to interpret our present findings on a structural basis in the context of the recently determined TRAP architecture, in which individual TRAP subunits were assigned positions within the overall density of the mammalian TRAP-complex in native ER membranes (Fig. 1a)[27]. As visualized by cryo-electron tomography, TRAPγ assumes a central position in the mammalian TRAP-complex, contacting eL38 and short rRNA expansion segment on the ribosome, thus coordinating the remaining TRAP subunits with the ribosome and the other translocon components: the Sec61-complex (contacted by TRAPαβ) and OST (contacted by TRAPδ). The bacterial ribosomal components uL24 and H59, both in vicinity to eL38 and TRAPγ, were recently observed to coordinate the SP for SRP binding[42]. Assuming a similar SP position in the mammalian system, the N-terminal SP tip may consequently be close enough to interact with eL38 and the cytosolic domain of TRAPγ during the hand-over of the SP from SRP to Sec61 (Fig. 1a). According to this hypothetical scenario, TRAP may support the insertion of SP into the Sec61-channel in the productive hairpin (rather than head-first) configuration.

We further propose that high GP content and low hydrophobicity may extend the time that SP dwell on the cytosolic surface of the Sec61-channel, and that TRAP can compensate for this by stabilizing SP on the cytosolic side and by potentially facilitating Sec61-channel gating on the lumenal side. This raises the question of how TRAP signals the presence of an SP-bearing ribosome to the Sec61-channel, and supports Sec61-channel gating. In vitro experiments support the concept that auxiliary components, such as TRAP, facilitate Sec61-channel opening in a substrate-specific manner, i.e., for precursor polypeptides with weak signal peptides[15,17,23]. Strikingly, the ER-lumenal domains of the TRAPαβ-subcomplex contact loop 5 in the hinge region between the N- and C-terminal halves of Sec61α, and thus may facilitate Sec61-channel opening to allow initiation of protein translocation[27]. This would be consistent with our live-cell $Ca^{2+}$ imaging experiments (Fig. 7).

Taken together, one-third of all polypeptides in a human cell are transported into or across the ER membrane via the Sec61-channel. While the Sec61-complex facilitates translocation of all polypeptides with amino-terminal SP or TMH the Sec61-associated TRAP-complex supports translocation in a substrate-specific manner. Our results suggest that TRAP may act as a receptor for precursor polypeptides with high GP SP (and TMH) on the ER membrane's cytosolic face and relays information to the ER lumenal hinge of Sec61α, thus assisting high GP SP in opening the Sec61-channel for protein translocation. This raises the interesting possibility that SP with high GP content in human cells allow TRAP-regulated access of a subset of precursor polypeptides to the Sec61-channel. Notably, TRAPα was found to be subject to phosphorylation and $Ca^{2+}$-binding[43]. Thus, either one or both of these two modifications appear as good candidates for TRAP- and, therefore, ER protein import-regulation. Therefore, future work will address the question if the intracellular distribution of TRAP clients with a dual intracellular location, such as Calreticulin, is affected by TRAP modification.

## Methods
**Materials.** SuperSignal West Pico Chemiluminescence Susbtrate (# 34078) was purchased from Pierce™, Thermo Fisher Scientific. ECL™ Plex goat anti-rabbit

**Table 1 Primary structures of signal peptides of putative TRAP clients**

| UniProt ID | Gene name | Signal peptide | TM | N-glyc | GP % | Hph |
|---|---|---|---|---|---|---|
| Q9UM22 * | EPDR1 | MPGRAPLRTVPGALGAWLLGGLWAWTLCGLCSLGAVG | − | + | 29.7 | 0.727 |
| Q9H6X2 * | ANTXR1 | MATAERRALGIGFQWLSLATLVLICAG | + | + | 15.6 | 0.262 |
| Q8IWB1 | ITPRIP | MAMGLFRVCLVVVTA | − | + | 6.7 | 1.951 |
| O00622 | CYR61 | MSSRIARALALVVTLLHLTRLALS | − | − | 0 | 1.017 |
| P54802 | NAGLU | MEAVAVAAAVGVLLLAGAGGAAG | − | + | 21.7 | 1.461 |
| Q9GZX9 | TWSG1 | MKLHYVAVLTLAILMFLTWLPESLS | − | + | 4.0 | 1.269 |
| Q13454 *** | TUSC3 *** | MGARGAPSRRRQAGRRLRYLPTGSFPFLLLLLLLLCIQLGGG | + | + | 24.4 | −0.01 |
| Q9Y3A6 */** | TMED5 | MGDKIWLPFPVLLLAALPPVLLPGAAG | + | − | 29.6 | 1.074 |
| Q9H0U3 | MAGT1 | MAARWRFWCVSVTMVVALLIVCDVPSASA | + | + | 3.5 | 1.227 |
| Q13214 | SEMA3B | MGRAGAAAVIPGLALLWAVGLGSA | − | + | 25.0 | 1.106 |
| Q9BRR6 * | ADPGK | MALWRGSAYAGFLALAVGCVFL | − | − | 13.6 | 1.356 |
| P02751 | FN1 | MLRGPGPGLLLLAVQCLGTAVPSTGA | − | + | 25.8 | 0.161 |
| P45877 */** | PPIC | MGPGPRLLLPLVLCVGLGALVFSSGAEG | − | + | 32.1 | 1.063 |
| Q9UMX5 | NENF | MVGPAPRRRLRPLAALALVLALAPGLPTARA | − | − | 22.6 | 0.281 |
| O14773 | TPP1 | MGLQACLLGLFALILSGKCSY | − | + | 15.8 | 1.526 |
| P15941 * | MUC1 | MTPGTQSPFFLLLLLTVLTVVTG | + | + | 17.4 | 1.310 |
| Q15582 | TGFBI | MALFVRLLALALALGPAATLA | − | − | 8.7 | 1.729 |
| O75629 | CREG1 | MAGLSRGSARALLAALLASTLLALLVSPARG | − | + | 12.9 | 0.880 |
| Q9ULF5 * | SLC39A10 | MKVHMHTKFCLICLLTFIFHHCNHC | + | + | 0 | 0.707 |
| Q08380 | LGALS3BP | MTPPRLFWVWLLVAGTQG | − | + | 22.2 | 0.523 |
| P08069 * | IGF1R | MKSGSGGGSPTSLWGLLFLSAALSLWPTSG | + | + | 26.7 | 0.404 |
| P08572 * | COL4A2 | MGRDQRAVAGPALRRWLLLGTVTVGFLAQSVLA | − | + | 20.0 | −0.04 |
| Q8N2U0 * | TMEM256 | MAGPAAAFRRLGALSGAAALGFASYGAHG | + | + | 24.1 | 0.292 |
| Q9UBV2 * | SEL1L | MRVRIGLTLLLCAVLLSLASA | + | + | 4.8 | 1.651 |
| Q969V3 | NCLN | MLEEAGEVLENMLKASCLPLGFIVFLPAVLLLVAPPLPAADA | + | + | 16.7 | 1.038 |
| O14672 * | ADAM10 | MVLLRVLILLLSWAAGMG | + | + | 15.8 | 1.775 |
| P11117 */** | ACP2 | MAGKRSGWSRAALLQLLLGVNLVVMPPTRA | + | + | 16.7 | 0.304 |
| P06756 * | ITGAV | MAFPPRRRLRLGPRGLPLLLSGLLLPLCRA | + | + | 29.7 | 0.163 |
| Q12907 | LMAN2 | MAAEGWIWRWGWGRRCLGRPGLLGPGPGPTTPLFLLLLLGSVTA | + | + | 31.8 | 0.230 |
| P56937 | HSD17B7 | MRKVVLITGASSGIGLALCKRL | + | + | 19.1 | 1.733 |
| P39656 */** | DDOST *** | MGYFRCARAGSFGRRRKMEPSTAARAWALFWLLLPLLGAVCA | + | − | 14.3 | 0.115 |
| Q8TB61 | SLC35B2 | MDARWWAVVVLAAFPSLGAG | + | − | 23.8 | 0.627 |
| Q6PIU2 | NCEH1 | MRSSCVLLTALVALA | − | + | 14.3 | 1.617 |
| Q5JPE7 | NOMO2 | MLVGQGAGLLGPAVVTAAVVLLLSGVGPAHG | + | + | 29.0 | 1.259 |
| P08236 | GUSB | MARGSAVAWAALGPLLWGCALG | − | + | 22.7 | 0.883 |
| P00533 * | EGFR | MRPSGTAGAALLALLAALCPASRA | + | + | 16.7 | 0.634 |
| Q5VW38 | GPR107 | MAALAPVGSPASRGPRLAAGLRLLPMLGLLQLLAEPGLG | + | + | 28.2 | 0.538 |
| QBN129 * | CNPY4 | MGPVRLGILLFLFLAVHEAWA | − | + | 14.3 | 1.312 |

Amino acid sequences of signal peptides (SP) are shown together with protein accession number, gene name, presence of transmembrane domains (TM) or N-glycosylation sites (N-glyc) in the mature domain, GP content of SP in %, and SP hydrophobicity (Hph). Signal peptides are divided into N-terminal, hydrophobic, and C-terminal domains according to Phobius prediction (www.phobius.sbc.su.se). According to predictions with the TMHMM server 2.0 (www.cbs.dtu.dk/services/TMHMM/), nine of the precursor polypeptides with SP comprise one transmembrane region in their mature domain, seven of which are type I membrane proteins, i.e. expose their mature N-terminus to the ER lumen or extracellular space[56]; ten of the precursor polypeptides with SP comprise more than one transmembrane region in their mature domain. Thus, a total of nineteen TRAP dependent precursor polypeptides with cleavable SP, or 50%, are membrane proteins. Notably, PPIC was not listed in Supplementary Table 5 because UniProtKB does not name it as a precursor with SP. However, PPIC has SP according to SignalP 4.1 server (www.cbs.dtu.dk/services/SignalP/). *, accession numbers of proteins, which were also negatively affected by Sec61 complex depletion; **, validated proteins; ***, OST subunits. The *DDOST* gene codes for Ost48

IgG-Cy5 (PA45011, used dilution 1:1,000), and ECL$^{TM}$ Plex goat anti-mouse IgG-Cy3 conjugate (PA43009, used dilution 1:2,500) were purchased from GE Healthcare. Horseradish peroxidase coupled anti-rabbit IgG from goat (A 8275, used dilution 1:1,000) and horseradish peroxidase coupled anti-mouse IgG from goat (A 9044, used dilution 1:10,000) were from Sigma-Aldrich. We purchased murine monoclonal antibodies against β-actin (Sigma, A5441, used dilution 1:10,000), Ost48 (Santa Cruz Biotechnology, sc-74408, used dilution 1:1,000), and the *myc*DDK-tag (Origene, TA50011, used dilution 1:1,000), and rabbit antibodies against ACP2 (Thermo Fisher Scientific, PA5-29961, used dilution 1:500), PPIC (Abcam, ab184552, used dilution 1:1,000), TMED5 (Sigma-Aldrich, HPA050289, used dilution 1:250). Additional rabbit antibodies were raised against purified canine proteins (Calreticulin, used dilution 1:250; GRP94, used dilution 1:500; GRP170, used dilution 1:500), recombinant human protein (Sil1, used dilution 1:500), a peptide corresponding to amino acid residues 82–96 of human Dad1 plus an amino-terminal cysteine (CKADFQGISPERAFAD, used dilution 1:250), the carboxy-terminal peptides of human PPIB (14-mer, used dilution 1:1,000), Sec61α (14-mer, used dilution 1:250) and TRAPα (15-mer, used dilution 1:500) plus an amino-terminal cysteine, the amino-terminal peptides of human TRAPβ (15-mer, used dilution 1:500) and SRβ (13-mer, used dilution 1:500) plus a carboxy-terminal cysteine, and an internal peptide of SRα (aa 137–150, used dilution 1:250) plus a carboxy-terminal cysteine. Antibody against Stt3b (used dilution 1:500) was a kind gift from Stephen High (Manchester University, UK). Antibody quality was previously documented[44]. We note that the full scans of blots are shown in Supplementary Figs. 8–23. MG 132 and Tunicamycin were obtained from Calbiochem (# 474790, #654380).

**Cell manipulation and analysis.** Informed consent was obtained from the families of the individuals who provided fibroblasts for this study. The protocols were approved by the Internal Review Board of the Sanford Burnham Prebys Medical Discovery Institute, La Jolla CA, USA. Cells from an SSR3-CDG patient were provided by Dr. Charles Marques Lourenço. TRAPγ-deficient cells (CDG359), TRAPδ-deficient cells (CDG406), and control fibroblasts (GM0038, from the Coriell Institute) were cultured at 37 °C in a humidified environment with 5% CO$_2$, in Dulbecco's modified Eagle's medium (DMEM; Gibco$^{TM}$, Thermo Fisher Scientific) containing 1 g/l glucose with the addition of L-glutamine, sodium pyruvate (HyClone, GE Healthcare), 10% FBS (Sigma-Aldrich), and 1% penicillin and streptomycin (Gibco$^{TM}$, Thermo Fisher Scientific). All cell lines used in this study have been either acquired directly from patients or obtained from the Coriell Institute, which maintains strict verification of cell lines.

HeLa cells (DSM no. ACC 57) were obtained from the German Collection of Microorganisms and Cell Cultures, routinely tested for mycoplasma contamination by VenorGeM Mycoplasm Detection Kit (Biochrom AG, WVGM), and replaced every five years by a new batch. They were cultivated at 37 °C in a humidified environment with 5% CO$_2$, in DMEM with 10% fetal bovine serum (FBS; Sigma-Aldrich) and 1% penicillin and streptomycin. Cell growth was monitored using the Countess® Automated Cell Counter (Invitrogen) following the manufacturer's instructions.

For gene silencing, $5.2 \times 10^5$ HeLa cells were seeded per 6-cm culture plate, followed by incubation under normal culture conditions. For *TRAPB* silencing, the cells were transfected with a final concentration of 10 nM targeting siRNA (Supplementary Table 3) (Qiagen), or 20 nM *TRAPA* targeting siRNA, or 20 nM

**Table 2 Primary structures of transmembrane helices of putative TRAP clients without cleavable signal sequences**

| UniProt ID | Gene name | Most N-terminal transmembrane helix | N-glyc | GP % | Hph |
|---|---|---|---|---|---|
| O15121 | DEGS1 | 33IKSLMKPDPNLIWIIIMMVLTQLGAFYIVKKDLDWKWVIF | − | 9.5 | 1.801 |
| Q99519 * | NEU1 | 10LPDRRWGPRILGFWGGCRVWVFAAIFLLLSLAASWSKAENDFG | + | 17.0 | 0.193 |
| P61803 */** | DAD1 *** | 19STPQRLKLLDAYLLYILLTGALQFGYCLLVGTFPFNSFLSGFI | − | 14.3 | 1.662 |
| P04920 * | SLC4A2 | 699DFRDALDPQCLAAVIFIYFAALSPAITFGGLLGEKTQDLIGVS | + | 16.7 | 1.791 |
| P41221 | WNT5A | 15GMAGSAMSSKFFLVALAIFFSFAQVVIEANSWWSLGMNNPVQM | + | 14.3 | 1.135 |
| Q68CQ7 | GLT8D1 | 1MSFRKVNIIILVLAVALFLLVLHHNFLSLSSLLR | + | 0 | 2.386 |
| Q15629 | TRAM1 | 20LQNHADIVSCVAMVFLLGLMFEITAKASIIFVTLQYNVTLPAT | + | 4.8 | 1.888 |
| P55061 * | TMBIM6 | 20TPSTQQHLKKVYASFALCMFVAAAGAYVHMVTHFIQAG | − | 4.8 | 1.531 |
| Q5T9L3 * | WLS | 3GAIIENMSTKKLCIVGGILLVFQIIAFLVGGLIAPGPTTAVSY | − | 19.1 | 1.230 |
| Q8TCJ2 */** | STT3B *** | 55AGLSGGLSQPAGWQSLLSFTILFLAWLAGFSSRLFAVIRF | + | 14.3 | 1.225 |
| Q6UW68 | TMEM205 | 5GNLGGLIKMVHLLVLSGAWGMQMWVTFVSGFLLFRSLPRHTFG | − | 14.3 | 1.385 |
| Q643R3 | LPCAT4 | 30HLSRLQRVKFCLLGALLAPIRVLLAFIVLFLLWPFAWLQVAGL | + | 8.7 | 2.387 |
| P35610 | SOAT1 | 130DELLEVDHIRTIYHMFIALLILFILSTLVVDYIDEGRLVLEFS | − | 0 | 2.542 |
| Q9UIQ6 * | LNPEP | 102ACSVPSARTMVVCAFVIVVAVSVIMVIYLLPRCTFTKEGC | + | 0 | 3.057 |
| P11166 | SLC2A1 | 2EPSSKKLTGRLMLAVGGAVLGSLQFGYNTGVINAPQKVIEEFY | + | 22.7 | 1.196 |
| Q8TCT9 | HM13 | 22TTRPPSTPEGIALAYGSLLLMALLPIFFGALRSVRCARGKNAS | + | 14.3 | 2.025 |
| Q15005 * | SPCS2 | 70EKYKYVENFGLIDGRLTICTISCFFAIVALIVWDYMHPFPESKP | − | 4.8 | 1.516 |
| Q8NHP6 * | MOSPD2 | 483KLEDQVQRCIWFQQLLLSLTMLLLAFVTSFFYLLYS | − | 0 | 2.133 |
| Q9NW15 | ANO10 | 197IDSIRGYFGETIALYFGFLEYFTFALIPMAVIGLPYYLFVWED | − | 9.5 | 1.736 |
| A0PJW6 | TMEM223 | 34VLLFEHDRGRFFTILGLFCAGQGVFWASMAVAAVSRPPVPV | − | 14.3 | 1.531 |
| P08962 | CD63 | 2AVEGGMKCVKFLLYVLLLAFCACAVGLIAVGVGAQLVLSQT | + | 9.52 | 2.491 |
| Q9BZH6 | WDR11 | 1117CSPQVNQKSKALLVLLSLGCFFSVAETLHSMRYFDRAALFV | − | 4.76 | 1.627 |

Amino acid sequences of N-terminal transmembrane helices (TMH) plus flanking regions are shown together with their positions in the full protein sequences, along with the protein accession number, gene name, presence of N-glycosylation sites (N-glyc), GP content in TMH in %, and TMH hydrophobicity (Hph). The most N-terminal transmembrane helices were identified according to the TMHMM server 2.0 (www.cbs.dtu.dk/services/TMHMM/). According to predictions with the same server, six of the precursor polypeptides without cleavable sp comprise only the shown single transmembrane helix, five of these are type II membrane proteins, i.e. expose their N-terminus to the cytosol[56]; the other sixteen of the precursor polypeptides without cleavable sp comprise more than one transmembrane region. *, accession numbers of proteins, which were also negatively affected by Sec61 complex depletion; **, validated proteins; ***, OST subunits

**Table 3 Primary structures of signal peptides of possible TRAP clients and non-TRAP clients**

| UniProt ID | Gene name | Signal peptide | TM | N-glyc | GP % | Hph |
|---|---|---|---|---|---|---|
| Q9H173 */** | SIL1 | MAPQSLPSSRMAPLGMLLGLLMAACFTFCLS | − | + | 16.1 | 0.992 |
| P27797 */** | CALR | MLLSVPLLLGLLGLAVA | − | + | 17.6 | 2.307 |
| P14625 * | HSP90B1 | MRALWVLGLCCVLLTFGSVRA | − | + | 9.52 | 1.446 |
| Q9Y4L1 * | HYOU1 | MADKVRRQRPRRRVCWALVAVLLADLLALSDT | − | + | 3.12 | −0.101 |
| P23284 * | PPIB | MLRLSERNMKVLLAAALIAGSVFFLLLPGPSAA | − | + | 12.1 | 0.961 |
| — | mut-PPIB | MLRLGPRNMKVLLPPALIAGSVFFLLLPGPSAA | − | + | 24.2 | 0.814 |
| Q9Y3A6 */** | TMED5 | MGDKIWLPFPVLLLAALPPVLLPGAAG | + | − | 29.6 | 1.074 |
| — | mut-TMED5 | MGDKIWLPFPVLLLAALPPVLLAAAAG | + | − | 22.2 | 1.250 |

Amino acid sequences of signal peptides (SP) are shown together with protein accession number, gene name, presence of transmembrane domains (TM) or N-glycosylation sites (N-glyc), GP content in %, and hydrophobicity (Hph). Signal peptides are divided into N-terminal, hydrophobic, and C-terminal domains according to Phobius prediction (www.phobius.sbc.su.se). *, accession numbers of proteins, which were also negatively affected by Sec61 complex depletion; **, validated proteins. TMED5 was copied from Table 1 for comparison. Mutated variants of TMED5 and PPIB were generated by quick change mutagenesis in one and two steps, respectively, and verified by sequence analysis. The CALR genes codes for Calreticulin, the HSP90B1 gene for Grp94, the HYOU1 gene for Grp170. mut, mutated

AllStars Negative Control siRNA (Qiagen) using HiPerFect Reagent (Qiagen) following the manufacturer's instructions. After 24 h, the medium was changed and the cells were transfected a second time. SEC61A1 silencing was performed similarly with 20 nM targeting siRNA (Supplementray Fig. 3)[15]. Silencing efficiencies were evaluated by western blot analysis using the appropriate antibodies and an anti-β-actin antibody from mouse. Primary antibodies were visualized with ECL^TM Plex goat anti-rabbit IgG-Cy5 or ECL^TM Plex goat anti-mouse IgG-Cy3 conjugate using the Typhoon-Trio imaging system combined with Image Quant TL software 7.0 (GE Healthcare). Alternatively (for ACP2, Dad1, Sec61α, Sil1, TMED5), peroxidase coupled anti-rabbit IgG or peroxidase coupled anti-mouse IgG (for mycDDK-tag) were employed in combination with SuperSignal West Pico Chemiluminescent Substrate and the Fusion SL (peqlab) luminescence imaging system with accompanying software.

To rescue the phenotype after TRAPB silencing the corresponding human cDNA, additionally coding for a carboxy terminal mycDDK-tag, was obtained in pCMV6-entry-vector (Origene, RC213580). Cells were treated with TRAPB-UTR siRNA as described above for 96 h. Six hours after the second transfection, the siRNA-treated cells were transfected with either vector or expression plasmid using Fugene HD (Promega).

For plasmid driven over-production of model precursor polypeptides, HeLa cells were cultured in the presence of siRNA for a total of 96 h. After 72 h, the cells were transfected with the respective pCMV6-entry-vector (Origene, RC203180, RC201143, RC229667, MR202609), coding for an additional carboxy terminal

mycDDK-tag, or with pcAGGSM2-SIL1-IRES-GFP using Fugene HD. After 73 h Tunicamycin (2 μg/ml) and/or after 88 h MG 132 (10 μM) were added where indicated. The primers which were used for quick change mutagenesis (Thermo Fisher Scientific) of the model precursor expression plasmids are given in Supplementary Table 4. The mutations were confirmed by DNA sequencing.

**Label-free quantitative proteomics.** Cells ($1 \times 10^6$) were harvested, washed twice in PBS, and lysed in buffer containing 6 M GnHCl, 20 mM tris(2-carboxyethyl) phosphine (TCEP; Pierce^TM, Thermo Fisher Scientific), 40 mM 2-chloroacetamide (CAA; Sigma-Aldrich) in 100 mM Tris, pH 8.0. The lysate was heated to 95 °C for 2 min, and then sonicated in a Bioruptor sonicator (Diagenode) at the maximum power setting for 10 cycles of 30 s each. The entire process of heating and sonication was repeated once, and then the sample was diluted 10-fold with digestion buffer (25 mM Tris, pH 8, with 10% acetonitrile). Protein extracts were digested for 4 h with endoproteinase lysC, followed by the addition of trypsin for overnight digestion. The next day, booster digestion was performed using an additional dose of trypsin. After digestion, peptides were purified via SDB-RPS StageTips[45], eluted as either one or three fractions, and loaded for mass spectrometry analysis. Purified samples were loaded onto a 50-cm column (inner diameter: 75 microns; packed with 1.9-micron beads) via the autosampler of the Thermo Easy nLC 1000 (Thermo Fisher Scientific). Using the nanoelectrospray interface, eluting peptides were directly sprayed onto the benchtop Orbitrap mass spectrometer Q Exactive HF (Thermo Fisher Scientific)[46]. The mass spectrometer was operated in a

data-dependent mode with survey scans from 300 to 1700 m/z, and up to 15 of the top precursors were selected and fragmented using higher energy collisional dissociation (HCD). Dynamic exclusion was enabled to minimize repeated sequencing of the same precursor ions. Raw data were processed using the MaxQuant computational platform[47]. The peak list was searched against Human Uniprot databases, with an initial precursor and fragment tolerance of 4.5 ppm. Cysteine carbamidomethylation was set as the static modification, and methionine oxidation and N-terminal acetylation as variable modifications. The match between the run feature was enabled, and proteins were quantified across samples using the label-free quantification algorithm in MaxQuant[48] as label-free quantification (LFQ) intensities. Notably, LFQ intensities do not reflect true copy numbers because they depend not only on the amounts of the peptides but also on their ionization efficiencies; thus, they only served to compare abundances of the same protein in different samples[31,46–48].

**Data analysis**. Each MS experiment provided proteome-wide abundance data as LFQ intensities for three sample groups—one control (non-targeting siRNA treated) and two stimuli (down-regulation by two different targeting siRNAs directed against the same gene)—each having three data points. Supplementary Fig. 24 provides a detailed listing for the number of proteins detected in the two Sec61 depletion experiments and the three TRAP depletion experiments, respectively. Missing data points were generated by imputation, whereby we distinguished two cases. For completely missing proteins lacking any valid data points, imputed data points were randomly generated in the bottom tail of the whole proteomics distribution, following the strategy in the Perseus software (http://www.biochem.mpg.de/5111810/perseus)[49]. For proteins having at least one valid MS data point, missing data points were generated from the valid data points based on the local least squares (LLS) imputation method[50]. The validity of this approach is demonstrated in Supplementary Fig. 25 (also see Supplementary Note 1). Subsequent to data imputation, gene-based quantile normalization was applied to homogenize the abundance distributions of each protein with respect to statistical properties (Supplementary Fig. 26). To identify which proteins were affected by Sec61α and TRAPβ knockdown in siRNA-treated cells relative to the non-targeting (control) siRNA treated sample, we log2-transformed the ratio between siRNA and control siRNA samples, and performed two separate unpaired $t$-tests for each siRNA against the control siRNA sample. The $p$-values obtained by unpaired $t$-tests were corrected for multiple testing using a permutation false discovery rate (FDR) test. Proteins with an FDR-adjusted $p$-value (i.e. $q$-value) of below 5% were considered significantly affected by knock-down of the targeted proteins. The results from the two unpaired $t$-tests were then intersected for further analysis meaning that the abundance of all reported candidates was statistically significantly affected in both siRNA silencing experiments. All statistical analyses were performed using the R package SAM (http://www-stat-class.stanford.edu)[51].

Protein annotations of signal peptides, transmembrane regions, and N-glycosylation sites in humans and yeast were extracted from UniProtKB entries using custom scripts. The enrichment of functional Gene Ontology annotations (cellular components and biological processes) among the secondarily affected proteins was computed using the GOrilla package[52]. Using custom scripts, we computed the hydrophobicity score and glycine/proline (GP) content of SP and TMH sequences. A peptide's hydrophobicity score was assigned as the average hydrophobicity of its amino acids according to the Kyte-Doolittle propensity scale (averaged over the sequence length)[53]. GP content was calculated as the total fraction of glycine and proline in the respective sequence.

**Quantitative real-time PCR**. Total RNA was isolated from harvested cells using the RNA Blood Kit (Qiagen). Reverse transcription was performed using the SuperScript VILO cDNA Synthesis Kit (Invitrogen, Thermo Fisher Scientific) and the cDNA was purified using the PCR Purification Kit (Qiagen). TaqMan® Gene Expression Assays (Applied Biosystems, Thermo Fisher Scientific) were used to perform quantitative real-time PCR of TRAPA (Hs00162340_m1), TRAPB (Hs00162346_m1), SRPRA (Hs00162326_m1), SRPRB (Hs00253639_m1), PPIC (Hs00211349_m1), TMED5 (HS00211349_m1), and ACP2 (Hs00900682_m1) in a StepOne Plus 96-well system (Applied Biosystems, Thermo Fisher Scientific). The Δct-values were calculated using ACTB (Hs00357333_m1) as a standard, and the values were normalized based on control siRNA-treated cells.

**Live-cell $Ca^{2+}$ imaging**. HeLa cells were loaded with 4 μM Fura-2 AM (Molecular Probes, Thermo Fisher Scientific) in DMEM, and incubated for 45 min at 25 °C[40,41]. Then the cells were washed twice and incubated at room temperature in $Ca^{2+}$-free buffer (140 mM NaCl, 5 mM KCl, 1 mM $MgCl_2$, 0.5 mM EGTA, 10 mM glucose in 10 mM HEPES-KOH, pH 7.35). Where indicated, HeLa cells were treated with siRNA for 96 h prior to $Ca^{2+}$ imaging, and were treated with 1 μM Thapsigargin (Molecular Probes, Thermo Fisher Scientific). Ratiometric measurements were conducted for 7.5 or 12.5 min using an iMIC microscope and the polychromator V (Till Photonics), with alternating excitation at 340 and 380 nm and measurement of the fluorescence emitted at 510 nm. The microscope was equipped with a Fluar M27 lens with ×20 magnification and 0.75 numerical aperture (Carl Zeiss), and an iXon$^{EM}$+ camera (Andor Technology). Images containing 50–55 cells/frame were sampled every 3 s using TILLvisION software

(Till Photonics). Fura-2 signals were recorded as the F340/F380 ratio, where F340 and F380 correspond to the background-subtracted fluorescence intensities at 340 and 380 nm, respectively. Cytosolic $[Ca^{2+}]$ was estimated from ratio measurements using an established calibration method[54]. Data were analyzed using Excel 2007. $P$-values were determined using unpaired $t$-tests.

## Data availability

The mass spectrometry proteomics data have been deposited to the ProteomeXchange Consortium via the PRIDE[55] partner repository with the dataset identifier PXD008178. Source files for western blots were deposited at Mendeley Data under the DOI 10.17632/w8jv9ngnsk.1 [http://dx.doi.org/10.17632/w8jv9ngnsk.1]. All other data supporting the findings of this study are available from the corresponding authors on reasonable request.

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

## Acknowledgements

We thank Dr. Nagarjuna Nagaraj (Max-Planck-Institute of Biochemistry, Biochemistry core facility, Martinsried, Germany) for MS-analyses, Monika Lerner (Homburg) for technical assistance, Dr. Stephen High (Manchester University, UK) for kind gifts of antibodies, and Drs. Stephen High, Johannes Herrmann (Kaiserslautern, Germany), Maya Schuldiner (Rehovot, Israel), and Martin van der Laan (Homburg) for comments on the manuscript. J.D., R.Z., S.P., and F.F. were supported by the DFG (ZI234/13-1, FO716/4-1), R.S. by IRTG1830. H.H.F. was supported by grant R01DK99551 and the Rocket Fund.

## Author contributions

D.N. performed the MS data analysis under supervision by V.H. R.S. carried out the validation experiments under supervision by J.D. and R.Z. S.S. performed the qPCR and $Ca^{2+}$ imaging experiments. H.F. provided CDG patient fibroblasts. J.D. planned and supervised the sample generation for MS analysis. S.P., F.F., J.D., and R.Z. designed the study and wrote the manuscript together with S.L. and V.H. All authors discussed the results and the manuscript.

## Additional information

**Competing interests:** The authors declare no competing interests.

