## [Peer Review File · Nature Communications]

Reviewers' comments:

Reviewer #1 (Remarks to the Author):

It has been known for a long time that the Translocon-associated protein (TRAP) complex facilitates nascent polypeptide translocation across the Sec61 channel in a substrate-dependent manner. It was furthermore demonstrated that the efficiency with which a signal peptide (SP) gates the Sec61 channel is inversely correlated with its dependence on TRAP. Nevertheless, the molecular basis for the varying degree of channel gating efficiency and of TRAP-dependence of different SPs was obscure, so that identification of the distinguishing features that underlie the observed differences remained an important challenge. The authors of the submitted manuscript have addressed this problem with a global approach, by carrying out a proteomic analysis of cells silenced for the TRAP complex. As expected, they find that the collection of proteins that are downregulated by TRAP depletion is enriched in proteins of the exo-endocytic pathway, specifically in SP-bearing-, membrane- and glycoproteins. Analysis of the SPs of their hits revealed sequences enriched in gly and pro, suggesting that a low propensity for alpha helix formation underlies inefficient Sec61 gating and TRAP dependency.

The authors have tackled an important question with methodology that had not previously been applied to this problem. However, the study has a few weak points that would have to be amended to make it suitable for publication in Nature Communications.

1. The main problem with the study is that the evidence for the conclusion that the hits of the screen are bona fide TRAP clients is indirect, based on the diminution of the expression levels of the hits. This diminution could be an indirect effect of TRAP depletion, not caused by reduced translocation in the absence of TRAP. In going over the supplementary tables, I noticed that not all SP-containing precursors that are TRAP hits are also reduced by Sec61 depletion (this analysis would have been easier, had the authors provided the supplementary tables in Excel format). This is quite surprising: if these precursors are translocated inefficiently by Sec61 in the absence of TRAP, they should be affected also by Sec61 depletion; maybe they are particularly good Sec61 substrates, using the residual amount of the channel present after silencing. In this case, however, one would not expect them to be TRAP-independent. This discrepancy illustrates to me the difficulty in drawing mechanistic conclusions simply from the expression levels of proteins in the silenced cells. The authors should confirm their initial findings from the screen by directly demonstrating that a few of their high gly-pro hits are indeed dependent on TRAP for translocation, and that reduction of the gly-pro levels converts them into TRAP-independent substrates. This could easily be done in semi-permeabilized cells of CDG patients in comparison to controls; the authors are well-versed in these techniques.

2. The heat maps of extended Fig. 1, b and e, show large differences in putative clients of Sec61 and TRAP, obtained with each of the two different Sec61 and TRAP-directed siRNAs, with large groups of proteins behaving in the opposite way. How is this explained and how does it impact the authors' conclusions? Were the log₂ values of Supplementary Tables 3-6 averaged from such divergent results? Also, the meaning of "control" should be explained. In most of the figures, "control" means cells that were treated with non-targeting siRNA; this is presumably the case also for extended Fig. 1, b and e; however, on p. 29 of the manuscript, we read: "To identify which proteins were affected by Sec61 α and TRAP β knock-down in siRNA-treated cells, we log₂-transformed the ratio between siRNA and control samples". Since the heat maps of Extended Fig. 1 show us ratios of control samples, the meaning of "control" in the quoted sentence is presumably not referred to scrambled siRNA transfected cells; maybe to non-transfected ones? This should be explained.

Minor points:

1. In the volcano plots of Fig. 1, d,f, the colour coding should be explained in the legend.
2. In Fig. 5, the term SSR2 should be replaced with TRAPbeta for consistency.
3. The plots of yeast SP GP content display a remarkable oscillatory behaviour. This needs some comment.
4. Some typos need to be corrected: e.g., on p. 3, Fig. 1Aa instead of Fig. 1a; on p.9, Fig 5 is

referred to as Fig. 3; the list of references is split.

Reviewer #2 (Remarks to the Author):

Dear authors,

I read your manuscript and I like the design of your experiments to discover clients of the TRAP complex.

However, I have some questions regarding the data:

Major points:

- After Sec 61 depletion about 7000 proteins detected, after TRAP depletion about 8,500 proteins. Can you explain why you detect more than 1500 additional proteins after TRAP depletion?
- You mentioned that roughly 1/3 of the data is not reproducible between experiments (page 4 and page 6). I have the feeling that this value is quite high, especially for this type of experiment where you need to compare peptide intensities with reduced experimental noise. Could you please provide a plot of the errors between the experiments?
- You showed regulation of 257 / 824 proteins for TRAP or Sec61. Did you expect to detect 3 fold more proteins for Sec61? The ratio between up and down regulation ((482/342 and 180/ 77) of the experiments differ significantly. Is this expected?
- Extended data figure 2:
 - Did you detect differences in the results when you use different siRNA´s for the same protein?
 - In the manuscript you presented few examples of a validation of the results by western blot and qRT PCR. The data indicate that you validate 1% of the regulated proteins. Do you have examples where the validation did not work?
 - Statistical analysis of the data: On page 29 you describe imputation of data points. Did you do a statistical analysis if the number of data points is sufficient that this imputation will not change results?

Overall I have the feeling that the authors should repeat parts of the experiments to reduce the measurement error.

- a) In case you do label free quantification, please use spike in standards for normalization between runs or directly conduct a DIA experiment. This will also solve the problem with missing data points.
- b) Alternative to this: You could use TMT / iTRAQ for the experiments. This would also reduce noise and the missing data point problem.
- c) Please do a MS based validation using PRM/MRM (for this reduced number of proteins it should be possible also with SIL peptides for selected proteins as internal standard).
- d) Please provide copy number estimations for all proteins and define a cut off value for the noise.

Minor points:

- In the materials and methods parts many details should be added to enable readers of the manuscript to do similar experiments.
- Color code figure 3: on my screen / printout all green bars have the same color. Should be changed.

Reviewer #3 (Remarks to the Author):

In this study, Nguyen and coworkers combined siRNA-mediated TRAP depletion in Hela cells, label-free quantitative proteomics, and differential expression analysis to characterize TRAP-dependent clients. They validated three potential clients by western blotting and quantitative RT-PCR. By

analyzing signal peptides and transmembrane helices of TRAP clients, the authors found that TRAP clients have above-average glycine-plus-proline content, which could serve as a distinguishing feature. By live-cell Ca²⁺ imaging experiments, the authors suggest that TRAP may regulate the gating the Sec61 translocon channel during translocation or insertion of substrates. The key and novel finding from this paper is the identification of glycine-plus-proline content in TRAP dependent clients by mass spectrometry. However, the authors should perform additional experiments to support this conclusion.

Major comments:

1. In Figure 3, the authors need to include a few control Sec61a specific clients that do not depend on TRAP activity. It is possible that the subtle effects that they see with three substrates (TMED5, PPIC and ACP2) may not be specific for TRAP clients.

2. To determine if the high content of glycine and proline in signal sequences or transmembrane helices is a distinguishing feature of TRAP clients, the authors should mutate Glycine to Alanine in potential TRAP clients (TMED5, PPIC, and ACP2). They can then transfect these mutants into control or TRAP depleted cells to determine TRAP dependency. Conversely, the authors can also introduce a few glycine and proline residues into Sec61 specific clients to show that they are now dependent on TRAP as well. These experiments would strengthen the manuscript since the identification of the high content of glycine and proline in TRAP clients is the significant finding of the manuscript.

3. The authors claim that TRAP acts as a receptor on the membrane's cytosolic face for precursor polypeptides with high glycine and proline content. In order to make this claim, the authors need to perform interaction studies between TRAP and its clients including Sec61 clients as controls, either in cells or in vitro translated substrates. Since authors suggest that the high content glycine and proline may increase the dwelling time for substrates at the Sec61 translocon/TRAP complex, it is possible to capture the interaction by co-immunoprecipitations.

4. The Ca²⁺ releasing live image data is a series of indirect data to support the conclusion TRAP affects Sec6-channel gating. It is possible that the depletion of TRAP may have indirect effects such as interfering the expression of Ca²⁺ pumps in the ER.

Minor comments:

1. The whole manuscript is mainly focusing on the negatively affected proteins from the mass spectrum; maybe they should discuss briefly the positively affected proteins. It is clear from recent studies that the depletion of Sec61 activates IRE1-mediated UPR pathway.

2. In Fig. 3, the authors need to describe all abbreviations since it is not immediately clear what is co, G, D1, and D2.

3. Line 222 Fig. 3a, b / Line 224 Fig. 3c, d should be Fig 5 a, b, c, d.

4. The authors mentioned MG132 in their purchase list, but it is not clear where it was used.

Answers to the comments of the referees:

Reviewer #1

Thank you for your view that we have tackled an important question with methodology that has not previously been applied to this problem.

Weak points:

1. *The main problem with the study is that the evidence for the conclusion that the hits of the screen are bona fide TRAP clients is indirect, based on the diminution of the expression levels of the hits. This diminution could be an indirect effect of TRAP depletion, not caused by reduced translocation in the absence of TRAP. In going over the supplementary tables, I noticed that not all SP-containing precursors that are TRAP hits are also reduced by Sec61 depletion (this analysis would have been easier, had the authors provided the supplementary tables in Excel format). This is quite surprising: if these precursors are translocated inefficiently by Sec61 in the absence of TRAP, they should be affected also by Sec61 depletion; maybe they are particularly good Sec61 substrates, using the residual amount of the channel present after silencing. In this case, however, one would not expect them to be TRAP-independent. This discrepancy illustrates to me the difficulty in drawing mechanistic conclusions simply from the expression levels of proteins in the silenced cells. The authors should confirm their initial findings from the screen by directly demonstrating that a few of their high gly-pro hits are indeed dependent on TRAP for translocation, and that reduction of the gly-pro levels converts them into TRAP-independent substrates. This could easily be done in semi-permeabilized cells of CDG patients in comparison to controls; the authors are well-versed in these techniques.*

Response: Indeed, it is surprising that some precursors were affected by TRAP but not by Sec61 depletion. We believe this to be related to what you have pointed out in your introductory statement, that „the efficiency with which a signal peptide gates the Sec61 channel is inversely correlated with its dependence on TRAP“. Therefore, this group of precursors is more sensitive to TRAP depletion, after which there are many TRAP-free Sec61 channels left as opposed to the scenario of Sec61 depletion, where all Sec61 channels are TRAP associated. We have added a statement towards this end to the Introduction on page 3 and the Results on page 7.

In the original manuscript, we had included validation results for three putative TRAP clients (ACP2, PPIC, TMED5). In addition to this, we now extensively extended our validations by i) characterizing three more TRAP client candidates (the OST subunits Dad1, Ost48, Stt3b) by independent silencing and western blot experiments with TRAP β -targeting siRNAs and by complementation assays, plus two potential candidates (Sil1, Calreticulin), which were dismissed by statistical analysis because they were not affected by both TRAP-targeting siRNAs to the same extent, plus three un-affected ER proteins as negative controls (GRP94, GRP170, PPIB), and by ii) studying import of model precursors in the presence or absence of TRAP in the simultaneous absence or presence of MG132 (to inhibit the proteasome) after their plasmid driven expression in intact HeLa cells. The first part of this newly added work confirmed the three un-affected proteins as such and, thus, added three positively evaluated proteins to our list as well as two potential candidates, which are at the border between low and high GP content in their signal peptides and were only mildly affected according to proteomic and western blot analysis but showed a clear TRAP dependent phenotype in the complementation analysis. We have added these data to the Results on pages 8/9 and new Fig. 4.

In the second part, the model precursors included a pair of PPI paralogs. These show 72% sequence identity in their mature domains but differ in the GP content of their signal peptides (32 versus 16%). According to validation data in new Figs 3 and 4, the precursor of PPIC depends on TRAP for ER import, whereas PPIB does not. This was confirmed by studying import of three model precursors (PPIB, PPIC, Sil1) in the presence or absence of TRAP in the simultaneous absence or presence of MG132 (to inhibit the proteasome) after their plasmid driven expression in intact HeLa cells. In these experiments, a low level of mature protein (PPIC, Sil1 but not PPIB) in the absence of TRAP correlated with accumulation of precursor in the presence of MG132 but not in its absence. Thus, these experiments confirmed both our conclusion on the relevance of GP content in signal peptides for TRAP dependence and our starting hypothesis for the proteomic approach that precursors are degraded by the proteasome in the cytosol when their import into the ER is inhibited. The data was added to the Results on pages 10/11 and new Fig. 6.

Furthermore, we created a hybrid precursor comprising the sp of TRAP dependent PPIC and the mature part of TRAP independent PPIB, termed PPIC-PPIB, and found TRAP dependence to correlate with high GP content of the signal peptide. That is hybrid precursor PPIC-PPIB phenocopied PPIC, i.e., accumulated in the absence of Sec61- as well as TRAP-complex. We have added these data to the Results on pages 10/11 and new Fig. 6.

Finally, we subjected two control fibroblasts and three CDG patient fibroblasts with TRAP-deficiency to label-free quantitative proteomic analysis and differential protein abundance analysis and analyzed the data for negatively affected proteins, i.e. potential TRAP clients. As we describe on page 12 of the Results, 279 proteins were negatively affected by TRAP deficiency in the three patient fibroblasts versus the two control fibroblasts using the same analysis workflow as for the HeLa cells. Proteomic analysis confirmed the absence of TRAP complex in fibroblasts from CDG patients with mutations in the *TRAPG* or *TRAPD* genes²⁷ (Supplementary Table 8). 36% of the negatively affected proteins, i.e. 100 proteins were assigned to the secretory pathway, including 34 proteins with sp, 45% or 22 of which have sp with a GP content of > 10%. The average GP content of the sp of the negatively affected proteins was 15.8%. Thus, the results from these chronically TRAP-depleted cells also confirmed that the GP content of sp plays a dominant role for TRAP dependence of precursor polypeptides in ER protein import.

The CDG patient fibroblasts grow very slowly. Therefore, we refrained from doing the experiments in these cells. Furthermore, these cells had a long time to adapt to the absence of TRAP and can be expected to give a distorted view on the question of interest.

2. The heat maps of extended Fig. 1, b and e, show large differences in putative clients of Sec61 and TRAP, obtained with the each of the two different Sec61 and TRAP-directed siRNAs, with large groups of proteins behaving in the opposite way. How is this explained and how does it impact the authors' conclusions? Were the log₂ values of Supplementary Tables 3-6 averaged from such divergent results? Also, the meaning of "control" should be explained. In most of the figures, "control" means cells that were treated with non-targeting siRNA; this is presumably the case also for extended Fig. 1, b and e; however, on p. 29 of the manuscript, we read: "To identify which proteins were affected by Sec61 α and TRAP β knock-down in siRNA-treated cells, we log₂-transformed the ratio between siRNA and control samples". Since the heat maps of Extended Fig. 1 show us ratios of control samples, the meaning of "control" in the quoted sentence is presumably not referred to scrambled siRNA transfected cells; maybe to non-transfected ones? This should be explained.

Response: Throughout the manuscript, control means that cells were treated with non-targeting siRNA, the exception being control patient fibroblasts. We tried to be explicit about this throughout the manuscript.

As correctly noticed by the reviewer, the heat maps reflect the complexity of these analyses. To stay away from this complexity as much as possible, we focussed our analysis on those proteins, which were consistently affected by both siRNAs in the same direction. Moreover, the *p*-values obtained by *t*-tests were corrected for multiple testing using a permutation false discovery rate (FDR) test. Proteins with an FDR-adjusted *p*-value (i.e., *q*-value) of below 5% were considered significantly affected by knock-down of the targeted proteins. This is mentioned on page 5 and described in detail on page 18 of the revised manuscript.

Minor points:

1. The color coding of all figures follows the cryoelectron tomography structure of ER bound ribosomes of Fig. 1a and is explained in the corresponding figure legend.
2. Old Fig. 5 was changed as requested.
3. The human sp dataset comprises 3528, the yeast dataset only 830. The average length of human sp is 24.32 +/- 7.7 amino acid residues, of yeast 21.89 +/- 4.7. Thus, the first peak refers to no GP per yeast sp, the second to one GP, the third to two GPs, and so on. This causes the oscillatory appearance. This appears to be obscured by the higher variation in length of human sp. We have added a statement towards this end to the legend of new Fig. 5.
4. The typos and references were corrected.

Reviewer #2

Thank you for your kind words on the design of our experiments

Major points:

1. After Sec 61 depletion about 7000 proteins detected, after TRAP depletion about 8,500 proteins. Can you explain why you detect more than 1500 additional proteins after TRAP depletion?

Response: In the original manuscript, we reported approximated numbers, which were based on the first Sec61 experiment (6960 detected proteins) and the second TRAP experiment (8007 detected proteins). As suggested by this reviewer, we now report mean values with standard deviations computed from the individual replicates. Then, the number of proteins detected in Sec61 and TRAP silencing experiments is 7212 +/- 356 and 7670 +/- 332, respectively. The observed difference of about 460 is just a bit outside of the standard deviation and hence not statistically significant. We have updated the numbers in the Results on pages 4 and 6 accordingly.

The observed differences between samples (and replicates) may reflect biological variations of the samples and/or slight differences in siRNA silencing efficiencies and sample preparation for the MS analysis, and/or technical aspects of the MS instruments and MaxQuant settings/configurations.

The **attached Fig. 1** provides a detailed listing for the number of cases detected in the two Sec61 silencing experiments (two leftmost columns) and in the three TRAP experiments (three rightmost columns). The blue bars represent the proteins that we analyzed in this study. Marked in green are the proteins that do not have sufficient control data points, i.e. more than 2/3 of the control samples have missing data points. Marked in yellow are "contaminants" from MaxQuant analysis. These entries contain noisy data (indicated by ID started by "CON" or "REV") which should be ignored by the downstream analyses (see <https://www.nature.com/articles/nprot.2016.136> and <https://www.ncbi.nlm.nih.gov/pmc/articles/PMC4803341/>). Marked in red are proteins that cannot be found (or contain "invalid control") in other corresponding experiments. So far, we refrained from incorporating **attached Fig. 1** of this response letter into the Supplemental Information, but would be happy to do so if required.

2. You mentioned that roughly 1/3 of the data is not reproducible between experiments (page 4 and page 6). I have the feeling that this value is quite high, especially for this type of experiment where you need to compare peptide intensities with reduced experimental noise. Could you please provide a plot of the errors between the experiments?

Response: See our reply to the previous point. **Attached Fig. 1** illustrates the differences observed between replicates. Statistically, there are approximately 7486 (+/- 387) proteins detected in each experiment.

3. You showed regulation of 257 / 824 proteins for TRAP or Sec61. Did you expect to detect 3 fold more proteins for Sec61? The ratio between up and down regulation ((482/342 and 180/ 77) of the experiments differ significantly. Is this expected?

Response: In case of the negatively affected proteins, it makes perfect sense that 482 proteins were affected by Sec61 depletion and only 180 by TRAP depletion, since almost all proteins going through the secretory pathway, (roughly 30% of all proteins) involve Sec61 in their ER import while only a subset is expected to involve TRAP. This is discussed in the Results on pages 6/7. We had no expectations for the number of positively affected proteins, since this was not previously addressed.

4. *Extended data figure 2: Did you detect differences in the results when you use different siRNA's for the same protein?*

Response: Indeed, these kinds of differences were detected, e.g. for TRAP depletion effects on the two proteins Calreticulin (p -values for first siRNA 0.0003, for second siRNA 0.2902) and Sil1 (p -values for first siRNA 0.00016, for second siRNA 0,0684). These p -values were obtained by t -tests of a single siRNA experiment against the control sample. These two proteins were included into the additional validation experiments, which are reported in the Results on pages 8 and 9 and new Fig. 4. Also see our reply to point 5.

To be on the safe side, we focussed our analysis on those proteins, which were consistently affected by both siRNAs in the same direction. Furthermore, the p -values obtained by t -tests were corrected for multiple testing using a permutation false discovery rate (FDR) test. Proteins with an FDR-adjusted p -value (i.e., q -value) of below 5% were considered significantly affected by knock-down of the targeted proteins. This is mentioned on page 5 and described in detail on page 18 of the revised manuscript.

5. *In the manuscript you presented few examples of a validation of the results by western blot and qRT PCR. The data indicate that you validate 1% of the regulated proteins. Do you have examples where the validation did not work?*

Response: Originally, we positively validated three (ACP2, PPIC, TMED5) out of 38+22=60 potential TRAP clients. One validation attempt had failed because the commercial antibody did not work in our hands. Meanwhile, we extensively extended our validations by characterizing three more TRAP client candidates (the OST subunits Dad1, Ost48, Stt3b), plus two potential candidates (Sil1, Calreticulin), which were dismissed by statistical analysis because they were not affected by both TRAP-targeting siRNAs to the same extent, plus three un-affected ER proteins as negative controls. This work confirmed the three un-affected proteins as such and added three positively evaluated proteins to our list (bringing the total number of positively validated TRAP clients to 6, corresponding to 10%) as well as two potential candidates, which are at the border between low and high GP content in their signal peptides and were only mildly affected according to proteomic and western blot analysis but showed a clear TRAP dependence phenotype in the complementation analysis. We have added these data to the Results on pages 8 and 9 and new Fig. 4.

6. *Statistical analysis of the data: On page 29 you describe imputation of data points. Did you do a statistical analysis if the number of data points is sufficient that this imputation will not change results?*

Response: We thank the reviewer for raising this valid point. Based on the reasonable assumption that the reason for missing values is that they stem from "the bottom" of the distribution (see below), we now performed a systematic analysis of the potential effect of imputing missing values. As will be shown, the results derived from imputed data have a high correlation of 0.97 or 0.93 with the original data (assuming that the missing values stem either from the bottom 5th quartile or 10th quartile of the distribution).

The validation steps are summarized in **attached Fig. 2** of this response letter. Generally speaking, we introduced artificially missing data by randomly removing 10% of the (known) data points from the lower end of the distribution. Subsequently, these "missing" data points were imputed using the imputation method described in the manuscript. Then, a differential analysis was carried out on the imputed and the original data. Finally, we compared the results of the differential analysis of the imputed and original data to validate the reliability of the imputation method.

In detail, the validation method consisted of the following steps:

1. Data preparation: the first Sec61 silencing experiment was selected for the validation. Out of all protein entries, we selected only those proteins that are "complete", i.e. out of nine entries no entry was missing. In this experiment, this was the case for 5715 out of 6960 proteins. For simplicity, all protein intensities were converted into log₂ values.
2. Data removal: to reproduce the missing data of proteins that have intensity values below the detection limit of the MS equipment, we randomly removed 10% of the data points that are below a certain threshold. In other words, out of, say, 100 values that are below the threshold, 10 values were randomly removed. We tested two different thresholds (5% and 10% quantile of the overall distribution) (see **attached Fig. 3** of this response letter). For both thresholds, we repeated the removal 100 times. Therefore, in total, we generated 200 new datasets with artificially generated "missing" data.
3. Data imputation and differential analysis: the imputation and differential analysis were carried out exactly as was done in the manuscript. For the comparison in the later step, the differential analysis was also performed using the original data.
4. Correlation calculation: using the results of the previous steps, the significantly affected proteins were labelled as 1 (positively affected) and -1 (negatively affected) while the unaffected proteins were labelled 0. Afterwards, we computed the Pearson correlation coefficient between the results of the original data and of the imputed data. The overall correlation coefficients for the 5th and 10th quantile thresholds are 0.975 +/- 0.018 and 0.927 +/- 0.020, respectively. This validation shows that the imputation method is reliable since the results of the original data and the imputed data are highly correlated.

So far, we refrained from incorporating **attached Fig. 3** of this response letter into the Supplemental Information, but would be happy to do so if required.

7. Overall I have the feeling that the authors should repeat parts of the experiments to reduce the measurement error.

a) In case you do label free quantification, please use spike in standards for normalization between runs or directly conduct a DIA experiment. This will also solve the problem with missing data points.

b) Alternative to this: You could use TMT / iTRAQ for the experiments. This would also reduce noise and the missing data point problem.

c) Please do a MS based validation using PRM/MRM (for this reduced number of proteins it should be possible also with SIL peptides for selected proteins as internal standard).

d) Please provide copy number estimations for all proteins and define a cut off value for the noise.

Response: In our quantitative proteomic approach, proteins were quantified across samples using the label-free quantification algorithm in MaxQuant as label-free quantification (LFQ) intensities, as developed by Drs. Jürgen Cox and Matthias Mann at the Max Planck Institute of Biochemistry in Martinsried, Germany. We pointed out on pages 17/18 of the revised manuscript that LFQ intensities do not reflect true copy numbers because they depend not only on the amounts of the peptides but also on their ionization efficiencies; thus, they only served to compare abundances of the same protein in different samples. As mentioned on page 26 of the revised manuscript, the MS analyses were done by Dr. Nagarjuna Nagaraj, who was trained by M. Mann and runs the core facility of the Max Planck Institute of Biochemistry in Martinsried.

Minor points:

1. We extended the Methods on page 18 as requested.

2. We have worked on the colors to improve the quality of the figures.

Reviewer #3

Major comments:

1. *In Figure 3, the authors need to include a few control Sec61a specific clients that do not depend on TRAP activity. It is possible that the subtle effects that they see with three substrates (TMED5, PPIC and ACP2) may not be specific for TRAP clients.*

Response: We extensively extended our validations by characterizing three more TRAP client candidates (the OST subunits Dad1, Ost48, Stt3b) by independent silencing and western blot experiments with TRAP β -targeting siRNAs and by complementation assays, plus two potential candidates (Sil1, Calreticulin), which were dismissed by statistical analysis because they were not affected by both TRAP-targeting siRNAs to the same extent, plus three un-affected ER proteins as negative controls (GRP94, GRP170, PPIB). This work confirmed the three un-affected proteins as such and added three positively evaluated proteins to our list as well as two potential candidates, which are at the border between low and high GP content in their signal peptides and were only mildly affected according to proteomic and western blot analysis but showed a clear TRAP dependent phenotype in the complementation analysis. We have added these data to the Results on pages 8/9 and new Fig. 4.

2. *To determine if the high content of glycine and proline in signal sequences or transmembrane helices is a distinguishing feature of TRAP clients, the authors should mutate Glycine to Alanine in potential TRAP clients (TMED5, PPIC, and ACP2). They can then transfect these mutants into control or TRAP depleted cells to determine TRAP dependency. Conversely, the authors can also introduce a few glycine and proline residues into Sec61 specific clients to show that they are now dependent on TRAP as well. These experiments would strengthen the manuscript since the identification of the high content of glycine and proline in TRAP clients is the significant finding of the manuscript.*

Response: As requested, we extended our validation to the relevance of GP content by studying import of model two precursors (PPIC and Sil1) in the presence or absence of TRAP in the simultaneous absence or presence of MG132 (to inhibit the proteasome) after their plasmid driven expression in intact HeLa cells. The model precursors included a pair of PPI paralogs (PPIB and PPIC), which show 72% sequence identity in the mature domain but differ in the GP content of their signal peptides (32 versus 16%) and do or don't depend on TRAP. In the case of PPIC and Sil1, a low level of mature protein in the absence of TRAP correlated with accumulation of precursor in the presence of MG132 but not in its absence. Thus, these experiments confirmed both our conclusion on the relevance of GP content in signal peptides for TRAP dependence and our starting hypothesis for the proteomic approach that precursors are degraded by the proteasome in the cytosol when their import into the ER is inhibited. The data was added to the Results on pages 10/11 and new Fig. 6.

Furthermore, we created a hybrid precursor comprising the sp of TRAP dependent PPIC and the mature part of TRAP independent PPIB, termed PPIC-PPIB, and found TRAP dependence to correlate with high GP content of the signal peptides. That is hybrid precursor PPIC-PPIB phenocopied PPIC, i.e., accumulated in the absence of Sec61- as well as TRAP-complex. We have added these data to the Results on pages 10/11 and new Fig. 6.

Furthermore, we subjected two control fibroblasts and three CDG patient fibroblast lines with TRAP-deficiency to label-free quantitative proteomic analysis and differential protein abundance analysis and analyzed the data for negatively affected proteins, i.e. potential TRAP clients. As we describe on page 12 of the Results, 279 proteins were negatively affected by TRAP deficiency in the three patient fibroblasts versus the two control fibroblasts using the same analysis workflow as for the HeLa cells. Proteomic analysis confirmed the absence of TRAP complex in fibroblasts from CDG patients with mutations in the *TRAPG* or *TRAPD* genes²⁷ (Supplementary Table 8). 36% of the negatively affected proteins, i.e. 100 proteins were assigned to the secretory pathway, including 34 proteins with sp, 45% or 22 of which have sp with a GP content of > 10%. The average GP content of

the sp of the negatively affected proteins was 15.8%. Thus, the results from these chronically TRAP-depleted cells also confirmed that the GP content of sp plays a dominant role for TRAP dependence of precursor polypeptides in ER protein import.

3. *The authors claim that TRAP acts as a receptor on the membrane's cytosolic face for precursor polypeptides with high glycine and proline content. In order to make this claim, the authors need to perform interaction studies between TRAP and its clients including Sec61 clients as controls, either in cells or in vitro translated substrates. Since authors suggest that the high content glycine and proline may increase the dwelling time for substrates at the Sec61 translocon/TRAP complex, it is possible to capture the interaction by co-immunoprecipitations.*

Response: We toned down our wording on pages 13 and 14 of the Discussion to make it clear that this was just a suggestion, which is based on our current results in combination with our previous cryoelectron tomography (Pfeffer, S. *et al.* Dissecting the molecular organization of the translocon-associated complex. *Nat. Commun.* 8, 14516 (2017)) and Nenad Ban's work in the bacterial system (Jooma, A., Boehringer, D., Leibundgut, M. & Ban, N. Structures of the *E. coli* translating ribosome with SRP and its receptor and with the translocon. *Nat. Commun.* 7, 10471 (2016)).

4. *The Ca²⁺ releasing live image data is a series of indirect data to support the conclusion TRAP affects Sec6-channel gating. It is possible that the depletion of TRAP may have indirect effects such as interfering the expression of Ca²⁺ pumps in the ER.*

Response: In our previous work, we did not detect any effects of Sec61 depletion on SERCA (Lang, S. *et al.* Different effects of Sec61 α -, Sec62- and Sec63-depletion on transport of polypeptides into the endoplasmic reticulum of mammalian cells. *J. Cell Sci.* 125, 1958-1969 (2011) and added a statement towards this end to the Results on page 12. Furthermore, we omitted or toned down the relevant parts of the manuscript to down-play the role of the calcium imaging in the whole story (Abstract, page 2 and Discussion, page 13).

Minor comments:

1. We did not detect any sign of UPR activation in the 96 h of the experiments, i.e. there was no over-production of BiP, GRP94, and GRP170 and the term UPR did not come up in the GO term analysis of the positively affected proteins. We created a special section on positively affected proteins on pages 7/8 of the Results.
2. The legend to old Fig. 3 was improved as requested.
3. The respective section of the Results was corrected.
4. At the time, we were getting ahead of ourselves. None of the experiments of the original manuscript involved MG132. In the revised manuscript, MG132 was involved when we studied import of model precursors in the presence or absence of TRAP in the simultaneous absence or presence of MG132 (to inhibit the proteasome) after their plasmid driven expression in intact HeLa cells. These experiments confirmed both our conclusion on the relevance of GP content in signal peptides for TRAP dependence and our starting hypothesis for the proteomic approach that precursors are degraded by the proteasome in the cytosol when their import into the ER is inhibited. We have added these data to the Results on pages 10/11 and new Fig. 6.

Attached Figure 1: Numbers of proteins in the experiments

Attached Figure 2: Imputation validation flowchart

Attached Figure 3: The 5th and 10th quantile of the overall data distribution

Reviewers' comments:

Reviewer #1 (Remarks to the Author):

The revised version of the manuscript by Nguyen et al. is substantially improved compared to the original manuscript. Particularly important is the analysis of additional hits as well as of negative control proteins, and the transfection experiments, which validate the proteomic approach for the identification of TRAP clients. Some problems must, however, be addressed before the manuscript is suitable for publication in Nature Communications.

1. The transfection experiments illustrated in the new Fig. 6, by demonstrating the accumulation of the precursor forms of some of the hits in MG132-treated cells, nicely validate the proteomic approach to identify TRAP clients. The PPIC-PPIB chimera confirms that TRAP-dependency depends on the signal sequence. I disagree, however, with the authors' statement that "these short term expression experiments confirmed the crucial role of a high GP content in sp of TRAP-dependent precursor polypeptides" and that "This result of the sp exchange experiment confirmed our conclusion that TRAP dependence of precursor polypeptides in ER protein import is mainly driven by the signal peptide AND THAT THE GP CONTENT PLAYS THE DOMINANT ROLE" (p. 11). As suggested in my review of the original manuscript, to directly demonstrate the role of GP content, the authors should carry out mutagenesis experiments, reducing the GP content of TRAP clients, and viceversa increasing it in TRAP-independent substrates. As glycine and proline are not hydrophobic, it may be difficult to distinguish the specific effects of these two aminoacids on disfavouring alpha helix formation from simple reduction of the hydrophobicity of the signal peptide. While the global analysis of Fig. 5 suggests that the two features (GP content and hydrophobicity) are not correlated, it would be important to compare signal peptides designed to have similar hydrophobicity but differing gly-pro content.

2. In my first report, I expressed my confusion as to the meaning of "control" cells (second part of point 2). While it was clear to me that control cells were those treated with scrambled, non-targeting siRNAs, my problem was, and remains, in understanding previous extended Fig. 1b and e (Supplementary Fig. 1 in the revised version). In both these panels, the top six lines illustrate variations in the control sample. If values in samples are normalized to control (scrambled siRNA), what normalization process has been applied to these six lines? According to the described method, they have been normalized against themselves, therefore there should be no variation. Instead, there are groups of upregulated and downregulated proteins. This should be explained.

3. I find the explanation that the authors offer for the existence of TRAP-dependent and apparently Sec61-independent substrates unconvincing: the fact that the efficiency with which a signal peptide gates the Sec61 channel is inversely correlated with its dependence on TRAP does not per se explain why many TRAP-dependent substrates can get away with reduced levels of Sec61. As the authors explain in their rebuttal, under conditions of Sec61 depletion, the remaining Sec61 channels are associated with TRAP, whereas under conditions of TRAP depletion there are many TRAP-free Sec61 channels. Thus, the explanation that I would offer is that TRAP-dependent, Sec61-"independent" substrates have a higher than average affinity for Sec61-TRAP complexes, and thus use the residual translocons more efficiently than many Sec61-dependent, TRAP-independent substrates.

Other comments:

On p. 11, it is stated that PPIB's sp has 16% GP content. The correct value is given in Table 3 (12.1%). In that table, one Pro residue is not highlighted in red.

The authors have carried out a proteomic analysis con CDG patients' fibroblasts. To what extent to the hits overlap with those of TRAP-silenced HeLa cells?

In Fig. 4, the group of panels d,e,f should be moved a little downward from panels a,b,c. The way

the figure is laid out now, the four labels within grey rectangles (Sil1, GRP170, etc.) are equidistant between panels c and d, making it confusing to read the figure.

On p.9 of the Results section, the high GP content of Sil1 and Calreticulin, versus TRAP-independent clients is discussed. However, this concept has not yet been addressed in the Results section, making the reading somewhat nonlinear. The discussion of GP content should be entirely contained in the subsequent section (Characteristics of TRAP clients).

Reviewer #2 (Remarks to the Author):

Dear authors,

1. I do not understand figure 1. If only 5200/5900 proteins identified in all three experiments, why is the CV only about 300.
I still have the feeling that one third of the data is not reproducible. Are all regulated proteins detected in all experiments? Or do they also differ inbetween the experiments.
2. I asked for a strategy / additional data to overcome the limitatio of missing datapoints. You simply did not anwer this question. I strongly recommend to generate additional datasets.
3. Did you consider the reanalysis of the data according to the PSI guidelines?
4. Is there any strategy to validate more results?

Reviewer #3 (Remarks to the Author):

The authors have addressed all my concerns, and the manuscript has been carefully revised. Importantly, the authors have provided further experimental evidence supporting their conclusion that TRAP facilitates translocation of secretory proteins with signal peptides enriched of glycine and proline residues. However, I would ask authors to display western blot images in Figure 4 that show clear differences between control and TRAP depleted cells. I understand that the authors have displayed images from three independent experiments. However, the quantification numbers do not seem to reflect the actual bands on many of the blots. Replacing these blots with blots that show noticeable differences would better convince the reader. Also, they need to justify clearly in the text that why they do not see noticeable differences in many of their blots including Figure 3c (PPIC), but the numbers show otherwise. Notably, I see the numbers do not match well with the bands from following images.

- 1) In Figure 4a, STT3B blot, all lanes look similar, especially compare between control and Sec61 depleted. Also, why does the actin blot show an additional band that looks like OST48? It seems misplaced.
- 2) In Figure 4a, Calreticulin blot, compare control and TRAP-UTR-siRNA.
- 3) In Figure 4b, OST48 blot, compare control and TRAP depleted or plus TRAP complemented
- 4) In Figure 4d, GRP170 blot, compare control and Sec61 depleted plus complemented.

Reviewer #1

The revised version of the manuscript by Nguyen et al. is substantially improved compared to the original manuscript. Particularly important is the analysis of additional hits as well as of negative control proteins, and the transfection experiments, which validate the proteomic approach for the identification of TRAP clients. Some problems must, however, be addressed before the manuscript is suitable for publication in Nature Communications.

1. *The transfection experiments illustrated in the new Fig. 6, by demonstrating the accumulation of the precursor forms of some of the hits in MG132-treated cells, nicely validate the proteomic approach to identify TRAP clients. The PPIC-PPIB chimera confirms that TRAP-dependency depends on the signal sequence. I disagree, however, with the authors' statement that "these short term expression experiments confirmed the crucial role of a high GP content in sp of TRAP-dependent precursor polypeptides" and that "This result of the sp exchange experiment confirmed our conclusion that TRAP dependence of precursor polypeptides in ER protein import is mainly driven by the signal peptide AND THAT THE GP CONTENT PLAYS THE DOMINANT ROLE" (p. 11). As suggested in my review of the original manuscript, to directly demonstrate the role of GP content, the authors should carry out mutagenesis experiments, reducing the GP content of TRAP clients, and viceversa increasing it in TRAP-independent substrates. As glycine and proline are not hydrophobic, it may be difficult to distinguish the specific effects of these two aminoacids on disfavoured alpha helix formation from simple reduction of the hydrophobicity of the signal peptide. While the global analysis of Fig. 5 suggests that the two features (GP content and hydrophobicity) are not correlated, it would be important to compare signal peptides designed to have similar hydrophobicity but differing gly-pro content.*

To take the validation the requested step further, the GP-rich sp of a TRAP dependent precursor (TMED5) and the sp of a TRAP independent precursor (PPIB) were changed to the opposite GP values with as little as possible impact on sp hydrophobicity by quick change mutagenesis. The two mutated precursors, termed *mut*-TMED5 and *mut*-PPIB, respectively, decreased the GP content of the TMED5 sp from 29.6 to 22.2 and increased the GP of the PPIB sp from 12.1 to 24.2, while changing the hydrophobicity only from 1.074 to 1.250 and from 0.961 to 0.814 (Table 3). According to the protean prediction tool of the DNASTAR software (Lasergene 12), these mutations increased the helix propensity of the TMED5 sp and decreased the helix propensity of the PPIB sp (Supplementary Fig. 5b). These mutant variants were subjected to the short term expression analysis. While the exchange of a PG pair for two alanines in the case of the TMED5 sp turned the TRAP dependent precursor into a TRAP independent one, the simultaneous replacements of the dipeptide SE by GP plus two alanines by the dipeptide PP in the PPIB sp had the opposite effect (Fig. 6f-i, Supplementary Fig. 6f-h). Thus, the combination of sp mutagenesis and short-term expression experiments strongly supports a crucial role of a high GP content in sp of TRAP dependent precursor polypeptides.

2. *In my first report, I expressed my confusion as to the meaning of "control" cells (second part of point 2). While it was clear to me that control cells were those treated with scrambled, non-targeting siRNAs, my problem was, and remains, in understanding previous extended Fig. 1b and e (Supplementary Fig. 1 in the revised version). In both these panels, the top six lines illustrate variations in the control sample. If values in*

samples are normalized to control (scrambled siRNA), what normalization process has been applied to these six lines? According to the described method, they have been normalized against themselves, therefore there should be no variation. Instead, there are groups of upregulated and downregulated proteins. This should be explained.

The reviewer wonders that there is certain variability among the control samples in the Supplementary Fig. 1b and e. The aim of the protein-based quantile normalisation is NOT to homogenise the values of one protein for the control samples. Rather, this normalisation removes the systematic variation among different experimental iterations (two iterations of Sec61 silencing experiment, three iterations of TRAP silencing experiment). It ranks the raw data, computes the averages, and replaces the original values by the ranked averages. As a result, the distributions of one protein become statistically identical across all iterations (see accompanying Fig. 1 below).

Panel (a) contains raw MS data points for the SSR2 protein. Panel (b) contains normalized data points for SSR2. The variation left after normalisation reflects the biological variation between samples.

In panel (a), SSR2 levels of the controls (indices 1-3) are higher than both siRNAs in experiment 2 (red) and higher than the first siRNA in experiment 1 (blue). In the third experiment (green), the second siRNA (indices 7-9) induces lower levels than in the controls and the first siRNA. The same conclusions can be drawn from panel (b).

The benefit of the normalized values in panel (b) is that the blue, red, and green distributions contain identical values. Thus, one can now apply standard statistical tests to identify the significant differences. We have added these data as Supplementary Fig. 10.

3. I find the explanation that the authors offer for the existence of TRAP-dependent and apparently Sec61-independent substrates unconvincing: the fact that the efficiency with which a signal peptide gates the Sec61 channel is inversely correlated with its dependence on TRAP does not per se explain why many TRAP-dependent substrates can get away with reduced levels of Sec61. As the authors explain in their rebuttal, under conditions of Sec61 depletion, the remaining Sec61 channels are associated with TRAP, whereas under conditions of TRAP depletion there are many TRAP-free Sec61 channels. Thus, the explanation that I would offer is that TRAP-dependent, Sec61-"independent" substrates have a higher than average affinity for Sec61-TRAP complexes, and thus use the residual translocons more efficiently than many Sec61-dependent, TRAP-independent substrates.

Extended as suggested on page 7 of the results.

Other comments:

On p. 11, it is stated that PPIB's sp has 16% GP content. The correct value is given in Table 3 (12.1%). In that table, one Pro residue is not highlighted in red.

Corrected as suggested.

The authors have carried out a proteomic analysis con CDG patients' fibroblasts. To what extent to the hits overlap with those of TRAP-silenced HeLa cells?

We have added a new section to the Results on pages 13/14 and Supplementary Figs. 2b and 7 plus Supplementary Tables 8-11 towards this end.

In Fig. 4, the group of panels d,e,f should be moved a little downward from panels a,b,c. The way the figure is laid out now, the four labels within grey rectangles (Sil1, GRP170, etc.) are equidistant between panels c and d, making it confusing to read the figure.

Corrected as suggested.

On p.9 of the Results section, the high GP content of Sil1 and Calreticulin, versus TRAP-independent clients is discussed. However, this concept has not yet been addressed in the Results section, making the reading somewhat nonlinear. The discussion of GP content should be entirely contained in the subsequent section (Characteristics of TRAP clients).

Corrected as suggested.

Reviewer #2

1. I do not understand figure 1. If only 5200/5900 proteins identified in all three experiments, why is the CV only about 300. I still have the feeling that one third of the data is not reproducible. Are all regulated proteins detected in all experiments? Or do they also differ in between the experiments.

It is correct that a part of the data (proteins) is only detected in a fraction of the experiments. However, our study does not claim to detect all TRAP candidates. This is not possible on the basis of the available data. Instead, we apply a conservative statistical testing scheme where only those proteins are considered as putative TRAP clients that are significantly affected by both siRNAs.

There may be further TRAP clients which either have long life times in the cell, so that they cannot be sufficiently affected by 3-4 days of siRNA silencing, or for which one siRNA was not efficient, or for other reasons.

We will address the topic of imputing missing values under the next point.

2. I asked for a strategy / additional data to overcome the limitation of missing datapoints. You simply did not answer this question. I strongly recommend to generate additional datasets.

We agree that performing additional MS experiments is likely to improve the statistics. However, such experiments are very costly (€4000 for one experiment) and require long waiting times (2-3 months) in the queue of the core facility of the MPI of Biochemistry. Hence, this was not an option for us. Instead, we provided additional experimental data (Western blots and transport assays) to validate our findings.

In the previous response letter, we presented a validation of the data imputation scheme. We showed that the imputed values correlate with known data with $r \sim 0.93 - 0.97$. We have added these data as Supplementary Fig. 9.

3. Did you consider the reanalysis of the data according to the PSI guidelines?

No, we do not plan to do this. We have applied imputation and normalisation methods that are well-established in the field:

1) B.M. Bolstad, R.A. Irizarry, M. Astrand, and T.P. Speed. A comparison of normalization methods for high density oligonucleotide array data based on variance and bias. *Bioinformatics*, 19(2):185–193, 2003.

2) S. Tyanova, T. Temu, P. Sinitcyn, A. Carlson, M.Y. Hein, T. Geiger, M. Mann, and J. Cox. The Perseus computational platform for comprehensive analysis of (prote)omics data. *Nature Methods*, 13:731–740, 2016.

3) H. Kim, G.H. Golub, and H. Park. Missing value estimation for DNA microarray gene expression data: local least squares imputation. *Bioinformatics*, 21(2):187–198, 2005.

4. Is there any strategy to validate more results?

We are convinced that we have done quite a bit of validation and the other two reviewers seem to agree with us: In case of the transient TRAP depletion experiments in HeLa cells, the proteomic data were derived from three independent experiments with triplicates and two different TRAP-targeting siRNAs each. Thus in principle, we have already validated the proteomic analysis twice, albeit with the same methodology. But the methodology itself is well established at the core facility of the Max Planck Institute of Biochemistry in Martinsried, it does not need any further validation. i) As a first independent validation, we performed western blot analyses for 10% of the putative TRAP substrates in independent experiments and confirmed the substrates as such by this independent method (ACP2, PPIC, TMED5, Dad1, Ost48, Stt3b). Simultaneously, we validated three negative control proteins and confirmed them as such (GRP94, GRP170, PPIB). ii) As a second validation, we tested four of the confirmed TRAP substrates (EPDR1, PPIC, Sil1, TMED5) plus one negative control protein (PPIB) in yet another experimental setting, i.e. under conditions of transient over-production of the substrates, and confirmed not only the four substrates as such but also the accumulation of their corresponding precursor in the presence of proteasome inhibitor (MG132). iii) As a third validation, we created a „signal peptide swap“ variant between the PPIC sp and the PPIB mature region and confirmed that the signal peptide drives TRAP dependence and that the GP content of signal peptides may play an important role for TRAP dependence. iv) As a fourth validation, we created two variants with mutagenized signal peptides (PPIB, TMED5) and confirmed that helix propensity of signal peptides drive TRAP dependence and that GP content of signal peptides, and as a result helix propensity, indeed play an important role for TRAP dependence. v) Finally, we tried to confirm the results from transient TRAP depletion in HeLa cells with proteomic analyses for fibroblasts from human patients, suffering from TRAP deficiency and, thus, congenital disorder of glycosylation, and at least partially succeeded.

Reviewer #3

The authors have addressed all my concerns, and the manuscript has been carefully revised. Importantly, the authors have provided further experimental evidence supporting their conclusion that TRAP facilitates translocation of secretory proteins with signal peptides enriched of glycine and proline residues. However, I would ask authors to display western blot images in Figure 4 that show clear differences between control and TRAP depleted cells. I understand that the authors have displayed images from three independent experiments. However, the quantification numbers do not seem to reflect the actual bands on many of the blots. Replacing these blots with blots that show noticeable differences would better convince the reader. Also, they need to justify clearly in the text that why they do not see noticeable differences in many of their blots including Figure 3c (PPIC), but the numbers show otherwise. Notably, I see the numbers do not match well with the bands from following images.

For quantification of western blots the values for each band were normalized to the corresponding value for β -actin (our loading control), which we failed to describe in the figure legends. This contributed to the invisibility of differences. Therefore, we have replaced the problematic blots by blots with comparable β -actin signals. The figure legends were corrected.

1) In Figure 4a, STT3B blot, all lanes look similar, especially compare between control and Sec61 depleted. Also, why does the actin blot show an additional band that looks like OST48? It seems misplaced.

Corrected as suggested.

2) In Figure 4a, Calreticulin blot, compare control and TRAP-UTR-siRNA.

Corrected as suggested.

3) In Figure 4b, OST48 blot, compare control and TRAP depleted or plus TRAP complemented

Corrected as suggested.

4) In Figure 4d, GRP170 blot, compare control and Sec61 depleted plus complemented.

Corrected as suggested.

Figure 1: The SSR2 intensity profile across all experiments (red - 1st experiment, green - 2nd experiment, blue - 3rd experiment) before and after quantile normalisation. The horizontal axis indicates sample conditions: 1 to 3 - control, 4 to 6 – SSR2 silencing by 1st siRNA, 7 to 9 – SSR2 silencing by 2nd siRNA

REVIEWERS' COMMENTS:

Reviewer #1 (Remarks to the Author):

The authors have now addressed all my concerns, so the manuscript can, in the opinion of this reviewer, be accepted for publication in Nature Communications. In Supplementary Fig. 2b, could the authors please specify which oval refers to CDG cells and which one to HeLa cells

Responses to the issues raised by the reviewers:

Reviewer #1 (Remarks to the Author):

— In Supplementary Fig. 2b, could the authors please specify which oval refers to CDG cells and which one to HeLa cells

The requested information was included.